# Three-dimensional hierarchically porous MoS₂ foam as high-rate and stable lithium-ion battery anode

Xuan Wei [1], Chia-Ching Lin[2], Chuanwan Wu[3], Nadeem Qaiser [1], Yichen Cai[1], Ang-Yu Lu [4], Kai Qi [1], Jui-Han Fu[5], Yu-Hsiang Chiang[1], Zheng Yang[1], Lianhui Ding[6], Ola. S. Ali[6], Wei Xu[6], Wenli Zhang[7], Mohamed Ben Hassine[1], Jing Kong [4], Han-Yi Chen [2] ✉ & Vincent Tung [1,5] ✉

Architected materials that actively respond to external stimuli hold tantalizing prospects for applications in energy storage, wearable electronics, and bioengineering. Molybdenum disulfide, an excellent two-dimensional building block, is a promising candidate for lithium-ion battery anode. However, the stacked and brittle two-dimensional layered structure limits its rate capability and electrochemical stability. Here we report the dewetting-induced manufacturing of two-dimensional molybdenum disulfide nanosheets into a three-dimensional foam with a structural hierarchy across seven orders of magnitude. Our molybdenum disulfide foam provides an interpenetrating network for efficient charge transport, rapid ion diffusion, and mechanically resilient and chemically stable support for electrochemical reactions. These features induce a pseudocapacitive energy storage mechanism involving molybdenum redox reactions, confirmed by in-situ X-ray absorption near edge structure. The extraordinary electrochemical performance of molybdenum disulfide foam outperforms most reported molybdenum disulfide-based Lithium-ion battery anodes and state-of-the-art materials. This work opens promising inroads for various applications where special properties arise from hierarchical architecture.

Architected materials, e.g., materials with 3D architectures at the micro- and nanoscale, hold tantalizing prospects for widespread applications, ranging from photonic devices to energy storage and conversion systems, mechanical reinforcement, wearable electronics, and biomedical devices[1-7]. Molybdenum disulfide (MoS₂) which represent three-atom-thick 2D building blocks, is a promising candidate for 3D architectures. From a structural perspective, MoS₂ exhibits an intrinsic hierarchy of structure features, such as phase heterojunctions, grain boundaries between crystalline domains of sizes ranging from millimeters down to micrometers, dislocations at the nanoscale, and point defects such as S vacancies on the atomic scale[8,9]. These structural features coupled with the high theoretical

capacity, earth abundance, and ease of solution processability make MoS₂ an attractive candidate as a Lithium-ion battery (LIB) anode[10-16]. However, its stacked and brittle layered structure limits the lithium-ion (Li⁺) diffusion and electrochemical stability, and therefore, only a handful of studies have described the Li-ion storage properties of pure single- and multi-layered MoS₂ nanosheets.

Current manufacturing routes toward 3D MoS₂, however, usually give rise to simple geometries of mesoporous and fractal-like features that recur only within two orders of magnitude, thus preventing researchers from combining MoS₂'s intrinsically attractive features with desirable material properties that are extrinsic to them[17-19]. As a result, their electrochemical performances remain inferior to those of

BP, Si, Si-graphene, Si@C, or graphene benchmarks[20–24]. Indeed, the field of architected materials has been almost exclusively focused on metallic and inorganic materials, and the detailed mechanistic insights have shed light on many guidelines for inducing the "stronger-yet-ductile", "lightweight-and-flaw-tolerable", "electrochemically reconfigurable", and "brittle-to-ductile" transitions. Developing such an understanding for 3D architected $MoS_2$ and beyond, e.g., graphene, MXene, and other transitional metal dichalcogenides (TMDs) shall open promising inroads for various applications where special properties and functionalities arise from the deliberate, multi-scale architecting of 2D atomic crystals.

Here, we demonstrate a dewetting-induced manufacturing scheme that enables the deliberate structuring of 2D ce-$MoS_2$ sheets into 3D hierarchically organized entities with extended control over structure-property relationships, delivering greatly enhanced mechanical and electrochemical properties purely by rational design without the change in chemical compositions. Advanced imaging, theoretical modeling, and comprehensive spectroscopic characterizations collectively reveal the well-organized 3D topological frameworks with spatially connected vortical truss unit cells that can be directly and continuously printed on the target substrate (4-inch-wafer scale) with an overall thickness of >50 μm. These characteristics enabled the fabricated $MoS_2$ foam anode, exceeding expectation, to deliver lithium (Li)-ion charge storage capacity compared favorably to the state-of-the-art layered black phosphorus (BP)[20], Si-graphene[23], Si@C[24], silicon (Si)[25,26], and mesoporous graphene particle anodes[27]. The 3D architected $MoS_2$ foam holds prospects for applications that require a combination of high-power density and long cycling life. These include wearable and implantable electric devices. In addition, with emerging high-voltage cathode materials, such as $LiNi_{0.5}Mn_{1.5}O_4$ (working potential is ~4.9 V vs. Li/Li$^+$), 3D architected $MoS_2$ foam may find good use as the anode in a 3.6 V cell. Furthermore, the excellent rate performance of 3D architected $MoS_2$ foam makes it an ideal candidate for Li-ion hybrid capacitors, potentially providing higher power density than Li-ion batteries and higher energy density than supercapacitors.

## Results

### Formation of MoS₂ foam

Our manufacturing scheme is presented in Fig. 1a. Bulk $MoS_2$ powders (Supplementary Fig. 1a) are chemically exfoliated into ce-$MoS_2$ and then re-dispersed in a mixture of deionized water and isopropyl alcohol (DI-$H_2O$: IPA = 7:3, v/v) for electrohydrodynamic (EHD) printing[28,29]. There are two phases in the EHD printing, (a) adjusting EHD conditions for creating uniform ce-$MoS_2$ containing liquid droplets under high voltage, forming a thin film on a copper (Cu) substrate pre-heated at 200 °C; (b) evaporation of the solvent simultaneously regulates the randomly dispersed 2D ce-$MoS_2$ into an orderly 3D architected assembly. We obtain different morphologies under different EHD conditions (at a constant 0.75 kV cm$^{-1}$ voltage). As shown in Supplementary Fig. 1b, when the fluid is under a "jet mode" at a flow rate of 20 μL min$^{-1}$, ce-$MoS_2$ nanosheets restack and form the wrinkled films at 25 °C. Under a low flow rate of 5 μL min$^{-1}$, tiny droplets are jetted out, forming fluffy crumples at a high temperature of 200 °C. When the flow rate increases to 7 μL min$^{-1}$ and the substrate temperature rises to 200 °C, the EHD is under the "cone-jet mode", where the Coulombic repulsion between ions extracts the solution out from the nozzle to generate an axisymmetric Taylor cone. The electrostatically driven instability at liquid-air interfaces is capitalized to continuously generate electrostatically charged droplets with a narrow distribution of diameters down to ~150 nm[30]. These tiny droplets form a thin film on the substrate. After that, the solvent evaporation confines the dispersed ce-$MoS_2$ sheets to the area between drying patches, and the ce-$MoS_2$ sheets are self-assembled into an ordered porous pattern and then truss unit cells[31].

The emergence of truss unit cells depletes the solvents at the truss-air contact line. Successive ce-$MoS_2$ are continuously carried off the droplets at the truss-air contact line, thus enabling the uninterrupted production of new layers of rings and struts in a bottom-up manner (Fig. 1a). The formation of 3D architected $MoS_2$ foam driven by EHD-printing is similar to our previous report (ref. 28) but is quite independent due to the difference in morphological evolution—randomly distributed and discrete crumples with hierarchically strained conformational elements, including facets, folds, ridges, vertices, and wrinkles vs. spatially ordered and coherent foam with hierarchically porous structural features, such as vortical truss unit cell, nanopores, and struts, intertwined $MoS_2$ sheets, tears and holes on the basal plane, and S vacancies. More details of the EHD condition optimization are provided in Supplementary Fig. 1 and Supporting Information.

### Characterization of MoS₂ foam

This engineering feat is particularly appealing because of the ability to join the evaporation-like simplicity of geometric patterns to the complexity of hierarchical architectures in a scalable fashion[3]. The result is the formation of 3D $MoS_2$ foam comprises hierarchical features with sizes spanning seven orders of magnitude in length scale, from angstroms to tens of centimeters, schematically represented and experimentally observed in Fig. 1b–i. On the macroscale, $MoS_2$ foam distributes ubiquitously and uniformly (4-inch wafer), as shown in Fig. 1b, c. In Fig. 1d, e, the truss unit cell (diameter of 1–3 μm) comprises layers of alternating rings and struts tapering down toward the bottom of the Cu substrate. These nanoscale struts made of intertwined and folded ce-$MoS_2$ sheets (Fig. 1f, g) structurally interconnect between layers of concentric rings, forming vertically stacked and ring-shaped viaducts[14]. These viaducts naturally define abundant transverse pores (pore size in the range of 150–250 nm) on the sidewalls of these vortical truss unit cells, as revealed in Fig. 1f. A close-up view of the ce-$MoS_2$ sheet (Fig. 1h) shows a high density of tears and holes on the 5–20 nm, highlighted by the dotted white lines. Figure 1i features a high percentage of atomically resolved defects, such as S-vacancies, derived from the harsh lithium (Li) intercalation reaction in chemical exfoliation, observed through the high-angle annular dark-field (HAADF) and aberration-corrected scanning transmission electron microscopy (STEM). More structural details are shown in Supplementary Fig. 2a. In parallel, $MoS_2$ foam firmly attaches to the underlying Cu substrate and can only be free-standing by dissolving the Cu substrate in an ammonium persulfate solution (Supplementary Fig. 2b). The difficulty of peeling off $MoS_2$ foam underscores the strong adhesion that ensures the establishment of uninterrupted conductive pathways. Furthermore, a close examination of surface morphology from the backside of $MoS_2$ foam enables us to correlate the predictive model with experimental observation (Supplementary Fig. 2c).

Other characterizations, including binding energies from X-ray photoelectron spectroscopy (XPS), characteristic peaks from X-ray diffraction (XRD), show signatures from Raman spectra and energy dispersive X-ray spectroscopy (EDS) mapping of relevant elements in a single truss unit cell as well as within the 3D networks, again prove the structural continuity and chemical coherence of 3D architected $MoS_2$ foam (Supplementary Figs. 2d, e, and 3). Meanwhile, 3D architected $MoS_2$ foam is substantially strained (-1.75 ± 0.15% vs. 3.2 ± 0.37% of tensile strain in crumples, based on the redshift magnitudes of the Raman $E_{2g}$ and $A_{1g}$ peaks in Supplementary Fig. 2f) and displays a relatively higher electron density than that of the wrinkled counterpart. These results agree well with the previous reports and have profound implications on activating the 3D architected $MoS_2$ with the significantly decreased ion diffusion barrier with ~0.2 eV for Li and greatly improved conductivity of 4.66 S m$^{-1}$ (discussed in the later section) compared to that of pristine 2H-$MoS_2$ bulk (0.42 eV for

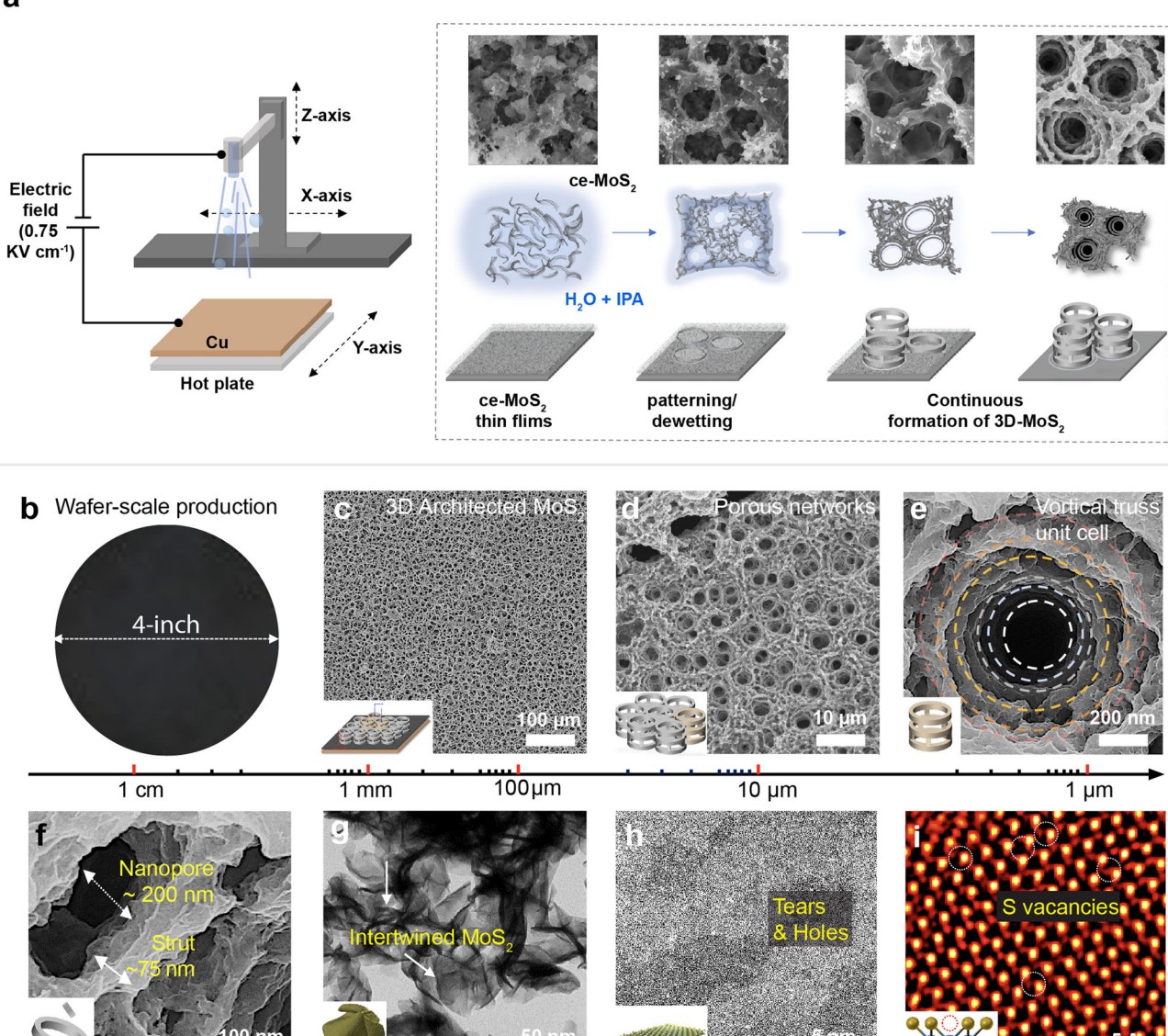

**Fig. 1 | Printing 3D MoS₂ foam through dewetting-induced manufacturing.**
**a** The manufacturing scheme illustrates the EHD setup and the structural evolution of the MoS₂ foam. **b** Demonstration of up-scalable manufacturing of MoS₂ foam on a 4-inch copper (Cu) substrate that comprises structural hierarchies over seven

orders of magnitude, including (**c**) interconnected porous networks, (**d**) architected structure, (**e**) vortical truss unit cell, (**f**) nanopores and struts, (**g**) intertwined MoS₂ sheets, (**h**) tears and holes on the basal plane, and (**i**) S vacancies.

Li-ion diffusion barrier and conductivity of 0.0576 S m⁻¹). In addition, strain-induced upshift of Mo $d$ states towards the Fermi level gives rise to a more robust interaction with metal ions, indicating that the storage capacity could be directly tailored at the atomic level. Consequently, the inherently strained structure of 3D architected MoS₂ foam opens inroads to manipulate the intrinsic activities of 2D MoS₂ building blocks, such as diffusion barrier, adsorption, and conductivity.

### Electrochemical performance

Indeed, the excellent electrochemical performance of MoS₂ foam validates the above analysis. MoS₂ foam anode demonstrates superior electrochemical performance to other reference samples, including crumples, wrinkled films, and bulk (Fig. 2a–c). In the rate capability test, the reversible specific capacity of MoS₂ foam is stabilized at 1575,

1550, 1515, 1431, 1268 and 1111 mAh g⁻¹ upon increasing the current density of 0.2, 0.5, 1.0, 2.0, 5.0, and 10 A g⁻¹, respectively (Fig. 2a). The Coulombic efficiency is >85% in the first cycle and >95% in the following cycles (Fig. 2b). Such high specific capacities, close to its theoretical capacity of 1675 mAh g⁻¹, especially at high current densities of 5 A g⁻¹ and 10 A g⁻¹, which is, to the best of our knowledge, beyond the previous MoS₂-based anodes and offers promise for meeting fast-charging requirements. When the current density is reduced to 0.2 A g⁻¹, a high specific capacity of 1525 mAh g⁻¹ is immediately resumed and stabilized there. Meanwhile, electrochemically driven dimensional changes in the anodes made of randomly restacked MoS₂ sheets lead to mechanical stress buildup at a charge-discharge current density of >5 A g⁻¹. The result is fatigue and capacity fading after only a few tenths of cycles. The corresponding galvanostatic charge/discharge profiles are shown in Supplementary Fig. 4a, b. It is,

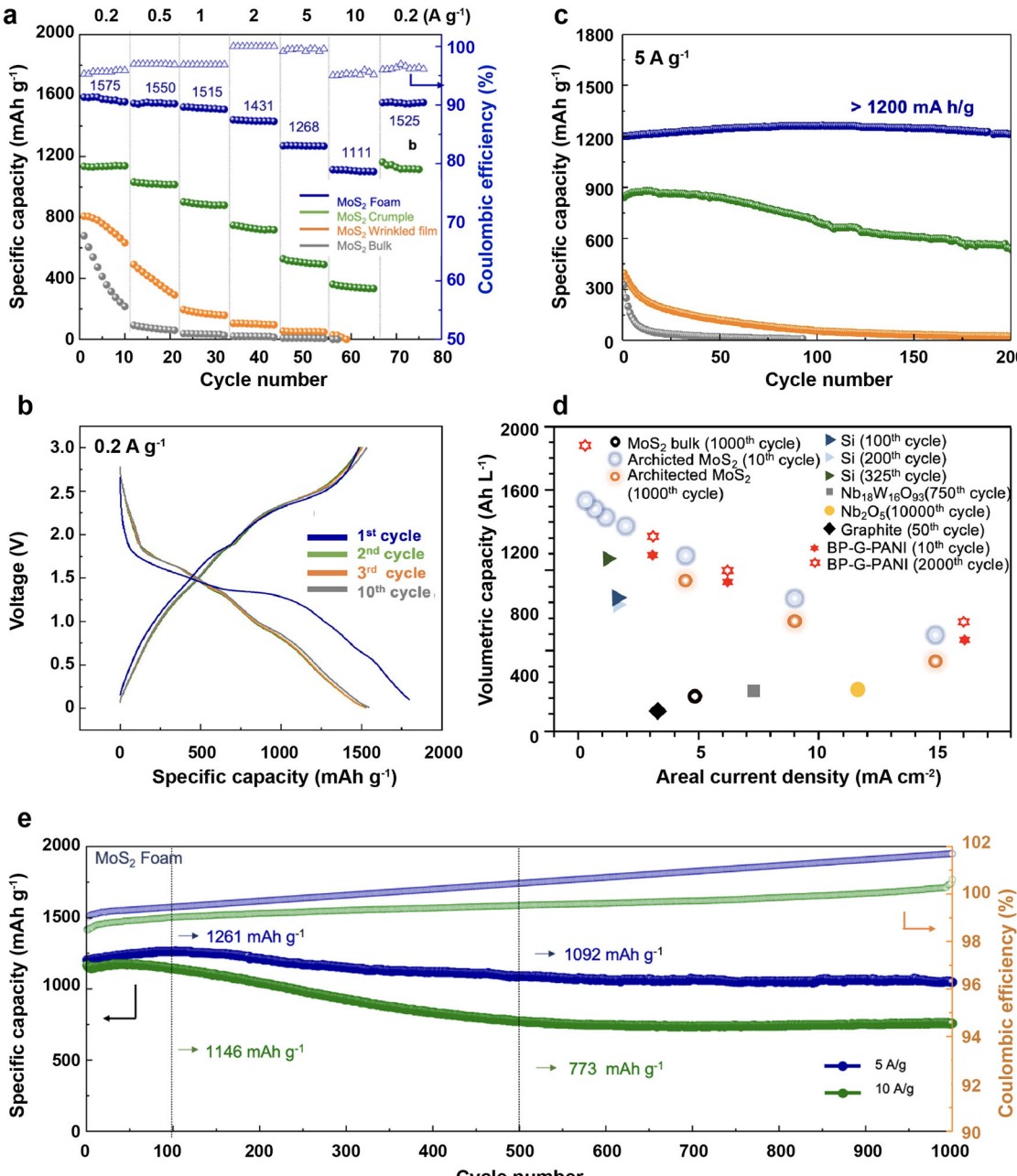

**Fig. 2 | Electrochemical performance of MoS₂ foam along with MoS₂ bulk, wrinkled film, and crumples. a** Rate capacity performance is measured at different current densities. **b** Galvanostatic discharge and charge profiles of MoS₂ foam were measured at the first 10 cycles. **c** Cycling stability comparison at the current density of 5 A g⁻¹. **d** The volumetric capacity of MoS₂ foam outperforms various state-of-the-art anodes and compares favorably to the current benchmark of 2D BP composite anodes, reproduced from ref. [20]. **e** Cycling performance of MoS₂ foam anodes was measured at current densities of 5 A g⁻¹ and 10 A g⁻¹ each for 1000 cycles.

therefore widely deemed impractical, and only a handful of studies have described the Li-ion storage properties of electrodes consisting of single- and multi-layered MoS₂ nanosheets.

Another advantageous feature of MoS₂ foam is the impressive electrochemical stability at a high current density beyond 5 A g⁻¹. In the static cycling test at the constant current density of 5 A g⁻¹ (Fig. 2c), the capacity of MoS₂ foam maintains 1200 mAh g⁻¹ in 200 cycles. In contrast, the capacity of the reference samples falters and decreases quickly after only 100 cycles. In the dynamic stability test (Supplementary Fig. 4c), the current density dynamically changed between 1 A g⁻¹ and 5 A g⁻¹ every ten cycles. The specific capacity of MoS₂ foam maintains >1500 and 1200 mAh g⁻¹ at 1 and 5 A g⁻¹, respectively, much higher and more stable than the reference samples. Moreover, the

MoS₂ foam delivered a reversible capacity of 1092 and 773 mAh g⁻¹ after cycling at 5 and 10 A g⁻¹ for 1000 cycles, respectively (Fig. 2e). Furthermore, continuously printing new layers of rings and struts enables us to prepare MoS₂ foam with a higher aerial mass loading. At the areal current density of 1 mA cm⁻², MoS₂ foam anodes with areal loadings of 1.0 and 2.2 mg cm⁻² perform stable for 150 cycles without noticeable degradation. Remarkably, when the areal loading is 3.5 mg cm⁻², the reversible areal capacity is 3.5 mAh cm⁻² and remains above 3 mAh cm⁻² after 100 cycles (Supplementary Fig. 4d). The electrochemical characterization of MoS₂ foam // LiFePO₄ full cell (weight ratio is MoS₂: LFP = 1:10) is shown in Supplementary Fig. 5. The anode is made of direct printing of pristine MoS₂ nanosheets into 3D hierarchical architecture on the targeted Cu substrates. Since no

binders or additives are used in the ink preparation, energy density is determined based on the total mass of printed $MoS_2$ foam (1 mg) and cathode LFP (10 mg). The energy density at $0.2 A g^{-1}$ is 0.0027 Wh / $(1 + 10)$ mg = 245 Wh kg$^{-1}$. Finally, by combining gravimetric capacity and electrode packing density $(1.05 \pm 0.05 g cm^{-3})$, we benchmark the performance of $MoS_2$ foam with the state-of-the-art, high-rate anodes made of conventional graphite, emerging BP, $Nb_2O_5$, Si, Si@C and reported $MoS_2$-based LIB anode (Fig. 2d, and Supplementary Tables S1–S3).

## Discussion

To understand how the hierarchical architecture enhances the capacity retention and rate capacity of $MoS_2$ foam as LIB anode, we conducted a combinatorial study of mechanical tests, theoretical modeling, and in-depth electrochemical analysis. It is widely known that the material's structural stability plays an important role in capacity retention. The electrochemically driven dimensional changes lead to mechanical stress buildup, ultimately resulting in fatigue and, thus, capacity fading after only a few tenths of cycles[32]. To this end, the structural stability of $MoS_2$ foam is firstly evaluated through mechanical tests. In the first test, $MoS_2$ foam with a thickness of 60 μm is compressed to 50% of its thickness (post-yield point). As shown in Fig. 3a, the failure in the foam is primarily localized in the densification of individually protruded vortical truss cells. At the same time, the overall structure remains intact in both vertical and horizontal directions, showing the global recovery. The post-yield deformation record is characterized by a ductile-like behavior with the continuous serrated flow[33], demonstrating the multiple-step structural deformation of the hierarchical structural elements instead of the one-step collapse of the whole structure (Fig. 3b). This behavior confirms the hierarchical architecture across multiple magnitudes. The several phases of structural deformation could be elastic ring buckling, shell buckling in individual struts, sliding between stacked sheets, and microcracking at nodes. Further, from an assembly perspective, the design of 3D architected foam consists of staggering 2D $MoS_2$ nanosheets with bond-free van der Waals (vdW) interfaces. These interfaces feature sliding and rotation degrees of freedom among the staggered nanosheets, endowing mechanical recovery and adaptability while retaining charge storage capability. Specifically, without the constraint of chemical bonding, such a 3D vdW architected foam offers a unique combination of mechanical resilience and response against electrochemical-mechanical fatigue. When deformed, the bond-free vdW interfaces enable 2D $MoS_2$ nanosheets to slide or rotate against each other, providing additional pathways to accommodate the continuously dynamic cycles of tension and compression. It is noted that while such vdW interfaces have recently been demonstrated in thin film formats to endow exceptional malleability and adaptability to irregular surface topographies, the 3D freestanding, additive-free architecture with vdW interfaces has not been reported elsewhere.

Indeed, 3D $MoS_2$ foam, instead of collapsing permanently after compression, recovered up to 95% of its original height after post-yield compression (50%), demonstrating impressive recoverability. In the second cyclic mechanical test, $MoS_2$ foam is compressed by 10% of its thickness (pre-yield). Impressively, $MoS_2$ foam rapidly recovers to 5% (Fig. 3c) along a straight line upon load releasing and shows remarkable resilience. The calculated elastic modulus is up to 2 GPa. As for references, the crystalline $MoS_2$ bulk and restacked wrinkled films with more condensed structures (Supplementary Fig. 6a, b) only slowly reverted from 10% to 7–8%, along with a curve behavior. The $MoS_2$ crumples (Supplementary Fig. 6c) are brittle and almost collapsed in the initial cycle under a small loading (<40 μN). These differences in mechanical adaptability underline the critical role of the 3D hierarchy in facilitating load dissipation and structural resilience, enabling the $MoS_2$ foam to overcome possible periodic structural deformation as a LIB anode.

## Numerical modeling and experimental results

Besides the mechanical tests, we further investigate the electrochemically driven dimensional changes in $MoS_2$ foam using the finite element methods (FEM) program, the COMSOL™ package. Here, the state-of-charge (SOC) is defined by the degree of lithiation, e.g., pristine or un-lithiated $MoS_2$ is defined as SOC of 0% while SOC of 100% is fully lithiated/charged $MoS_2$ foam. The boundary conditions and other details of simulation for electro-chem-mechanical numerical modeling for a battery electrode can be found in the method section[34]. Figure 3d features the spatial distribution of Li-ions at various stages of SOC. The corresponding volume expansion is about 70% at SOC of 100%, as recorded in Fig. 3e. From the FEM results, it becomes apparent that the spatially connected vortical truss unit cells help dissipate localized strain over the entirety of $MoS_2$ foam, limiting the volumetric expansion and thus preserving the structural integrity. In parallel, numerical calculations in Supplementary Fig. 7 show that the evolved von Mises stress distribution at a fully lithiated state (SOC of 100%) remains much lower (~few MPa) than its elastic modulus of 2 GPa, in the range of elastic deformation of $MoS_2$ foam (<1%), confirming the efficacy of foam structure[35]. SEM images featured in Fig. 3f–g also validate the simulation results. It is evident that the SEI layer forms a conformal coating along the supporting struts of $MoS_2$ foam after 1,000 charge/discharge cycles. Here, the large pore size and robust struts prevent aggregation and restacking of individual nanosheets, creating room for the SEI and, thus, the recoverability to remedy the volumetric expansion. Meanwhile, the combination of highly interconnected micro-channels and nano-channels ensures highly efficient ion transport throughout the entire network to reach the innermost pores, thus giving rise to continuous Li-ion diffusion and excellent reversibility.

In stark contrast, the SEI can be seen to rampantly deposit all over the reference samples (Supplementary Fig. 8). TEM, STEM, and corresponding EDX analysis of the SEI layer (Supplementary Fig. 9) further corroborate the observation in SEM. A thin layer of SEI (about 20 nm thickness) uniformly covers the contour of $MoS_2$ foam. The boundary between $MoS_2$ and SEI layer (represented by C and O) can be seen and determined by the corresponding EDX mapping and compositional analysis in Supplementary Table 4. Note that Li salts have been removed thoroughly, as evident by the vanished signal of P and F.

In addition, more characterizations, and mechanical tests of the post-cycling $MoS_2$ foam with SEI that are essential to understand the underlying mechanism further, are compiled in Supplementary Figs. 10–11. Anodes comprised of $MoS_2$ foam were pre-cycled (charge to 3 V vs. Li/Li$^+$) and then characterized by the Raman and XPS spectra. In Raman, we observed the broadening and redshift of both $E^1_{2g}$ and $A^1_g$ peaks, indicating pronounced strain and structural deformation. In parallel, XPS results substantiate our claim that the chemical composition and phase (trigonal prismatic, e.g., 2H) of $MoS_2$ foam remain unchanged after cycling. Also, we extended the post-characterization to assess the structural integrity and recoverability after cyclic deformation. As suggested in Fig. 3 and Supplementary Fig. 11, the 3D architected structure remains intact mainly by virtue of the great recoverability. Taken together, these post-cycling characterizations confirm the uniform SEI layer formation over the entire 3D architected $MoS_2$ foam and the impressive chemical and mechanical stabilities, ultimately resulting in much-improved capacity retention.

## Rate capability enhanced by improved intrinsic conductivity and Li$^+$ diffusion coefficients

Next, we analyze the intrinsic conductivity and electrochemical polarization of $MoS_2$ in bulk, wrinkled films, crumples, and foam through electrochemical impedance spectroscopy (EIS) and potentiostat intermittent titration technique (PITT) (Fig. 4a and Supplementary Figs. 12–13). The Nyquist plots of the impedance spectrum consist of two semicircles in the high-frequency region and a straight line in the

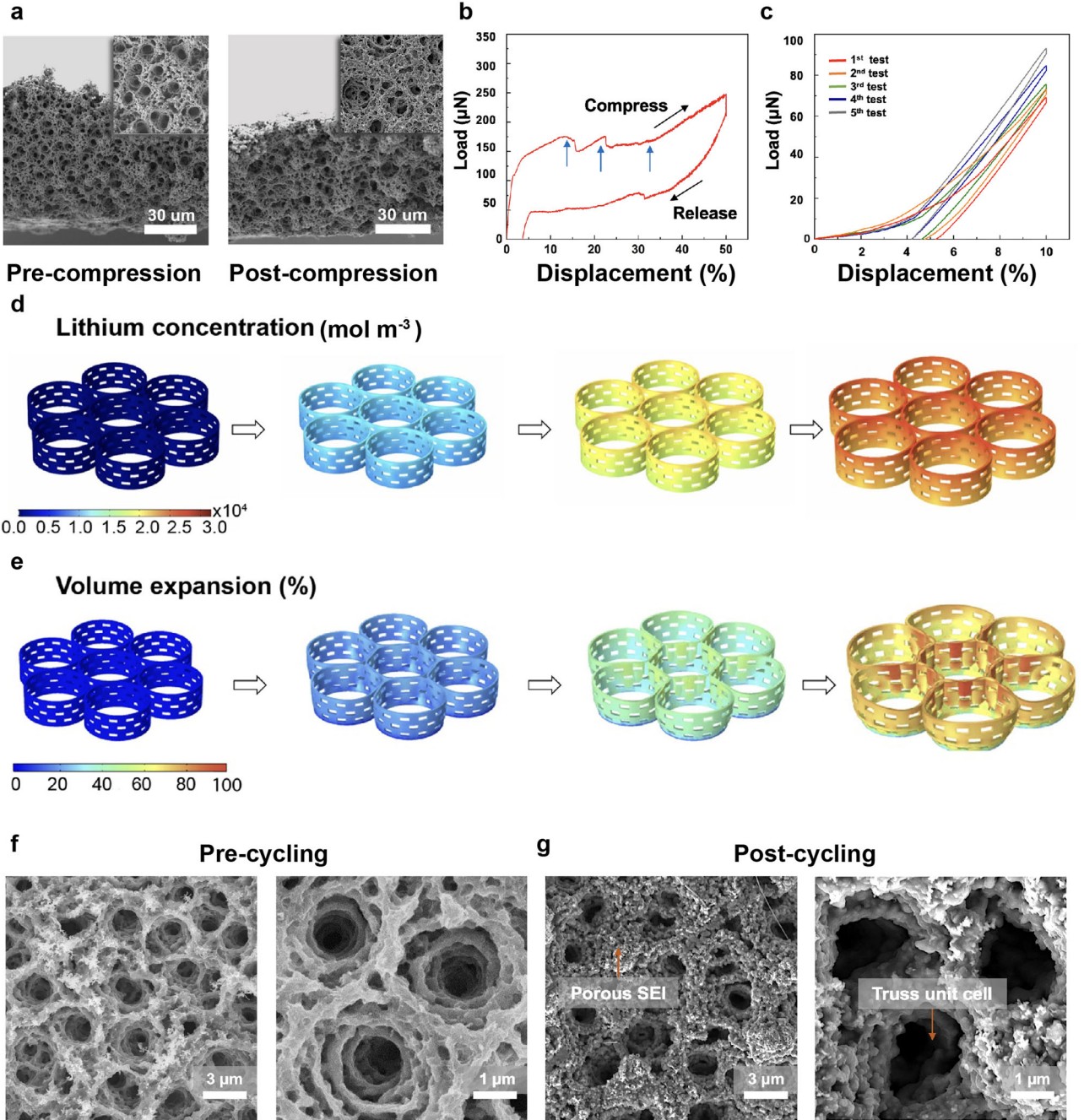

**Fig. 3 | Structural stability and capacity retention of MoS₂ foam. a** SEM images of pre- (left panel) and post-compression (right panel) of MoS₂ foam to 50% of displacement demonstrate an excellent recovery behavior. **b** The load and displacement curve (displacement to 50%) displays a ductile-like feature with continuous serrated flow (gray arrows), attesting to the multistep deformation of the hierarchical structure. **c** The load and displacement curve (displacement to 10%) exhibits a resilient feature with great recoverability. **d** Li-ion diffusion and the associated concentration distribution within the architected MoS₂ at different state-of-charge (SOC). The highest concentration of scale bar at SOC = 100% is calculated from the theoretical capacity of pristine MoS₂. **e** Volume expansion at various SOC. About 70% at SOC = 100%. SEM images of (**f**) pristine MoS₂ foam and (**g**) MoS₂ foam after 1000 charge/discharge cycles prove the uniform formation of the SEI layer while the hierarchical structure remains intact.

low-frequency region. The intercept of the semicircle with the Re (Z) axis in the high-frequency region presents the Ohmic resistance ($R_s$) of the entire cell; the diameters of the first and second semicircle are associated with the resistance of Li-ion migration through the solid electrolyte interface ($R_{SEI}$) and the charge-transfer resistance ($R_{ct}$), respectively; the slope of the straight line is related to ion diffusion efficiency[36]. As shown in Supplementary Fig. 12 and Table 5, the Nyquist plot demonstrates a decrease in both the series resistance and charge-transfer resistance for MoS₂ foam, while MoS₂ bulk and wrinkled films

exhibit an increase in these resistance values. Additionally, the straight-line slope for MoS₂ foam is steeper than the reference MoS₂ bulk at a low-frequency region, suggesting more surface capacitance and more efficient ion diffusion because of its shorter ion transmission paths. The result is the enhanced ion storage capacity and reaction kinetics[36]. Meanwhile, the well-organized foam structure significantly facilitates the formation of highly conformal and spatially distributed SEI layers (Fig. 3g) that guarantee electrochemical stability during charging and discharging. Meanwhile, the Li⁺ diffusion coefficients ($D_{Li}$) of MoS₂

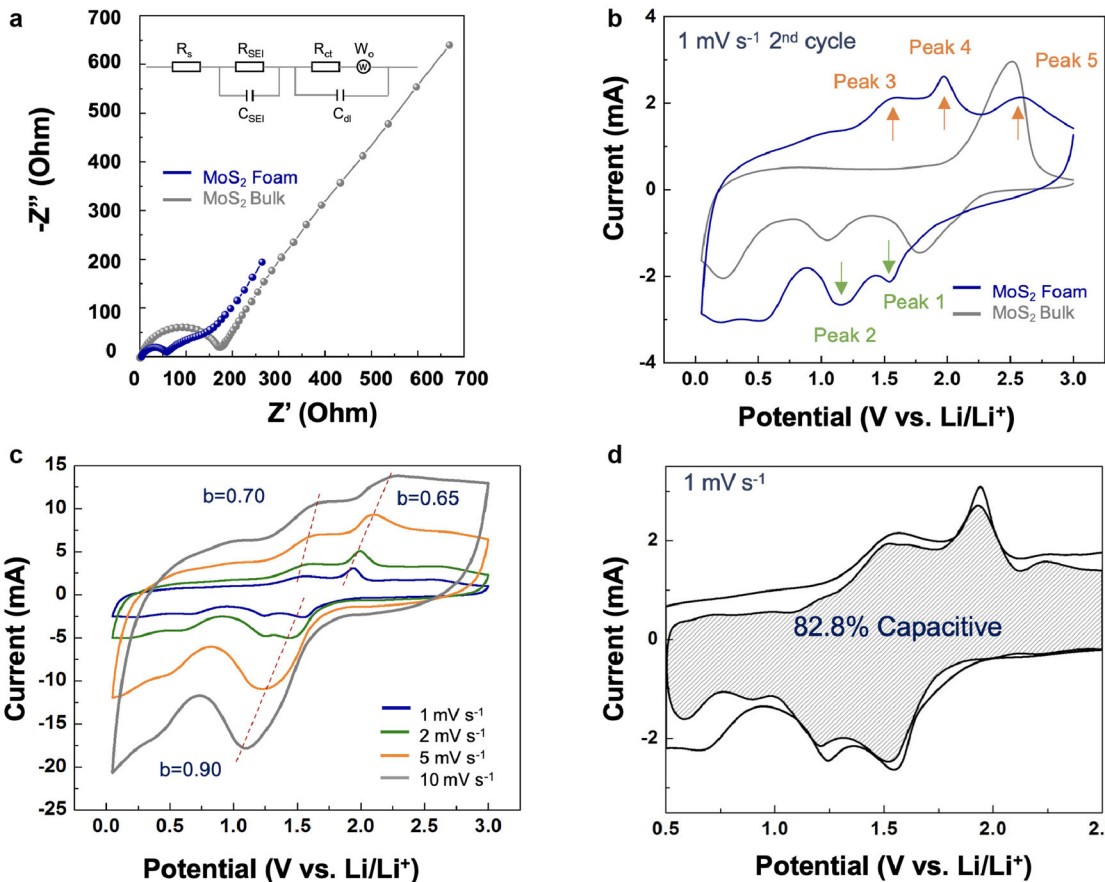

**Fig. 4 | Pseudocapacitive charge storage contributes to the MoS₂ foam's excellent rate capacity. a** Nyquist plots of MoS₂ foam and MoS₂ bulk electrode at a fully discharged state after ten cycles at 100 mA g⁻¹. **b** CV measurements feature the 2nd cycle of MoS₂ foam and MoS₂ bulk electrodes under 1 mV s⁻¹ in the voltage window between 0.01–3 V. **c** Capacitive effects are characterized by analyzing the CV curves at various sweep rates based on $i = av^b$, where the measured current $i$ follows a power-law relationship with the sweep rate $v$. **d** Capacitive and diffusion-controlled charge storage contributions for architected MoS₂ cycled in a Li-ion electrolyte at a scan rate of 1 mV s⁻¹.

foam, derived from the PITT in Supplementary Fig. 13, showed two to three orders of magnitude of enhancement than that of the reference MoS₂ bulk. These results again underscore the combination of manufacturing scalability, 3D hierarchically porous and spatially interconnected networks, multiscale architectural features, and strain-engineered ion diffusion barriers and conductivity suggests that 3D architected MoS₂ may be an ideal anode alternative with high-rate, high-capacity, high-mass-loading storage, and long-term cyclability. We have schematically correlated these appealing features with the formation of hierarchical structures within 3D architected MoS₂ as shown in Supplementary Fig. 14.

### Structure-induced pseudocapacitive charge storage involving different redox reaction pathways

The CV curves comparison between MoS₂ foam and reference MoS₂ bulk is shown in Supplementary Figs. 15 and 4b. The presence of the two pairs of redox peaks points to a different redox reaction pathway in MoS₂ foam. As indicated in Supplementary Figs. 15a and 4b, the first cycle of CV curves emanated from the MoS₂ bulk standard demonstrates the well-studied four-electron reduction reaction ($4Li^+$ + $MoS_2 + 4e^- \leftrightarrow Mo + 2Li_2S$) through an insertion (0.8 V vs. Li/Li⁺) and irreversible conversion mechanism (0.2 V). In contrast, the Li⁺ intercalation and Mo⁴⁺ reduction in 3D architected MoS₂ foam progress with at least three reductive steps. Similarly, the oxidative process in 3D architected MoS₂ foam occurs through discrete stages rather than the one-step Li₂S decomposition into S, Li⁺, and electrons ($Li_2S \leftrightarrow S + 2 Li^+ + 2e^-$ at 2.5 V vs. Li/Li⁺) in MoS₂ bulk[37,38]. We further conducted

Raman spectroscopy of MoS₂ foam before and after the first discharge cycle in tandem with MoS₂ foam after the first charge cycle (Supplementary Fig. 15b). We observed the formation of Li₂S after the 1ˢᵗ discharge cycle to 0.01 V vs. Li/Li⁺ and the emergence of sulfur after recharging to 3 V vs. Li/Li⁺. It is noted that the signature of Mo ions remains discernable, and the presence of LiOH is the result of the SEI formation process. In the following cycles (Fig. 4b), two highly reversible redox reaction couples (peaks 1&4 and 2&3) emerge and contribute most of the capacity. In contrast, peak 5 (2.5 V vs. Li/Li⁺) can be ascribed to Li₂S decomposition. We observed consistent trends in charge/discharge plots (Supplementary Fig. 4b), where the conversion reaction in MoS₂ foam gives rise to Li₂S and Mo. Notably, the resultant Mo atoms are in the vicinity (possibly remain in intimate contact) of the Li₂S matrix within the hierarchical foam structure, enabling rapid charge transfer with Li⁺ ions[36,39]. One Mo atom can accommodate up to six Li⁺ ions and then forms Mo/Liₓ clusters. Both Mo and S, therefore, participate in the reversible redox reactions, contributing to the very high specific capacity of >1500 mAh g⁻¹ at 1 A g⁻¹[39]. This highlights the presence of a different charge storage kinetics in hierarchically structured MoS₂ foam that is not observed in pristine MoS₂ bulk or stacked MoS₂ nanosheets.

To this end, we quantified capacitive and diffusion-controlled charge storage contributions from all samples, including the MoS₂ bulk standard, wrinkled films, crumples, and 3D architected foam, respectively. The capacitive contributions are quantified by collecting CV curves at 1, 2, 5, and 10 mV s⁻¹ over the potential range from 0.5 to 2.5 V vs. Li/Li⁺, based on the following equation: $i = k_1v + k_2v^{1/2}$,

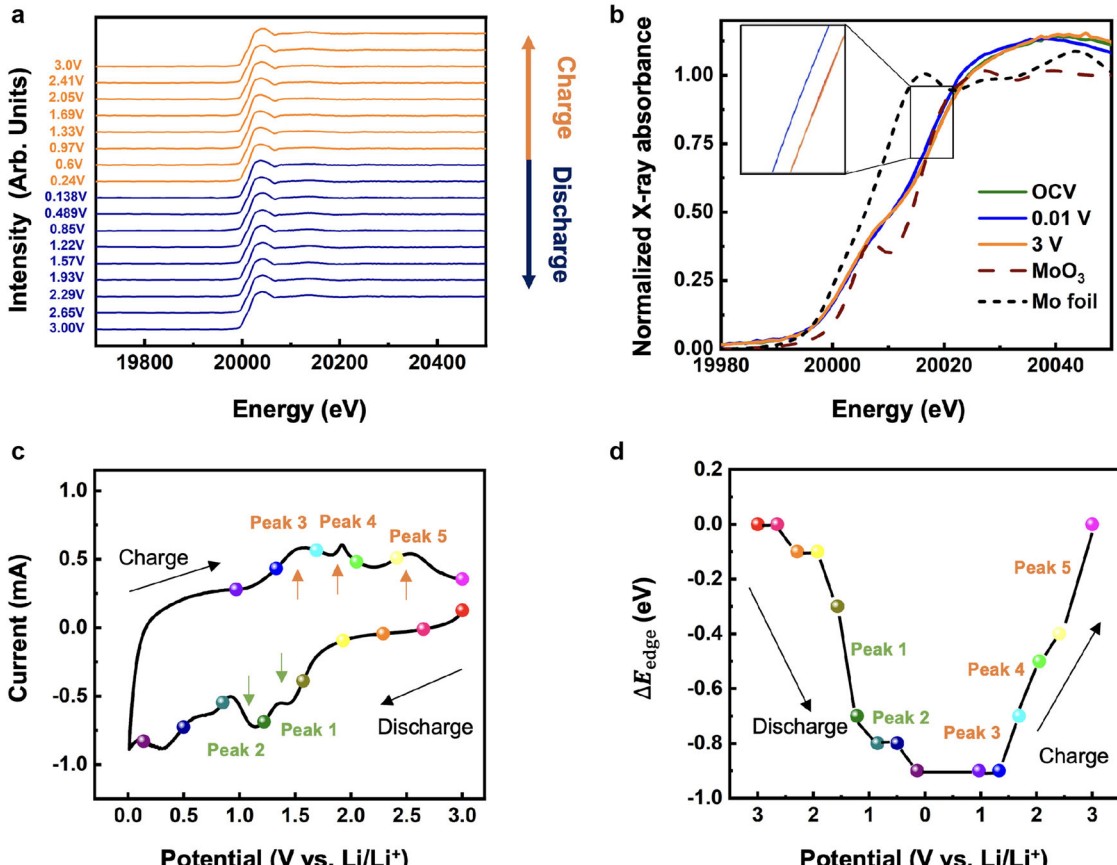

**Fig. 5 | In situ XANES tracking of Mo redox reaction in MoS₂ foam anode.**
**a** Normalized operando Mo K-edge XANES spectra of architected MoS₂ foam electrode measured at different potentials (blue: the discharging process; red: the charging process). **b** Normalized operando Mo K-edge XANES spectra for the first cycle at open-circuit voltage (OCV, 2.86 V), the first fully discharge (0.01 V), and the first fully charge (3 V) of MoS₂, compared with MoO₃ and Mo metal foil reference materials. **c** The cyclic voltammograms of architected MoS₂ foam anode at a scan rate of 0.3 mV s⁻¹ and (**d**) the corresponding absorption edge energy shift (ΔE$_{edge}$) of architected MoS₂ foam at different potentials (labeled as colored dots in the cyclic voltammograms).

where $k_1v$ and $k_2v^{1/2}$ correspond to capacitive and diffusion contributions to the measured current, respectively. $v$ is the scan rate (mV s⁻¹)[40–42]. Figure 4c, d, and Supplementary Fig. 16 summarize the results. The pseudocapacitive contributions for MoS₂ foam, crumples, wrinkled films, and bulk are 82.8%, 34.2%, 12.2%, and 0%, respectively, in good agreement with the calculated $b$-values. Accordingly, the capacitive behavior begins to take hold in wrinkled films because of increased surface area. The capacitive energy storage contribution becomes more pronounced in highly porous crumples because of the large surface area and highly defective basal plane. These properties are known to associate with oxidation and reduction reactions highly. Notably, in MoS₂ foam, the ratio of the pseudocapacitive storage to the total charge storage reaches 99% at 5 mV s⁻¹ (Supplementary Fig. 17). The predominate pseudocapacitive storage again highlights the unique combination of the easily accessible surface-redox active sites, the efficient interconnection, and the shorter charge-transfer distance from the surface to the innermost of the foam structure[40].

**Redox reaction of Mo ions**
We further reveal the energy storage mechanism in-depth. The spectroscopic characterization of *operando* Mo K-edge X-ray absorption near edge structure (XANES) corroborates the electrochemical redox reactions involved in the pseudocapacitive process. The XANES spectra for the second cycle with the potentials at an open-circuit voltage (OCV), the full lithiation (0.01 V vs. Li/Li⁺), the full delithiation (3 V vs. Li/Li⁺), and the references (MoO₃ and Mo metal foil) are displayed in

Fig. 5a, b. Metallic Mo is usually detected in the reported MoS₂ electrode as LIB anode, indicating the conversion reaction[41]. But in our MoS₂ foam, the absorption edge does not change to metallic Mo or Mo⁶⁺ through conversion reactions. Instead, the absorption edge of the MoS₂ foam anode slightly shifts to lower energy during the discharging (lithiation) process and then reverts during the charging (delithiation) process, suggesting that the redox reaction of Mo ions contributes part of the reversible capacity. However, the edge energy for all lithiation/delithiation spectra is within only c.a. 1 eV, which may be due to the anion–cation redox interactions between the Mo and S ions as observed in the literature[39,43], in agreement with our previous explanation. Figure 5c, d display the CV curve and the corresponding energy shift of the X-ray absorption edge ΔE$_{edge}$ during the second cycle, where ΔE$_{edge}$ is defined by the energy difference between the inflection points of OCV (c.a. 20018 eV) and certain potentials. It becomes clear that the edge energy decreases steeply from 2 V to 0.9 V vs. Li/Li⁺ vs. Li/Li⁺ during the discharging process (peaks 1 and 2) and increases sharply from 1 V to 2 V vs. Li/Li⁺ during the charging process (peaks 3 and 4). These potential ranges closely resemble the two pairs of redox peaks on the CV curve. Moreover, these peaks in the CV curve can still be recognized even at a high scan rate of 10 mV s⁻¹, revealing fast redox reactions (Fig. 4c). Therefore, the large capacity of 1500 mAh g⁻¹ is mainly the result of the reversible redox reactions of both S and Mo. The pseudocapacitive reactions of Mo ions mainly contribute to those redox couples, and the electrical double layer capacitive reactions contribute to the rest capacity. Therefore, the hierarchical structuring of 2D ce-MoS₂ into 3D architected MoS₂ foam

shall greatly enhance Li-ion electrochemical activity to near its theoretical value[36,42].

Finally, we note that the dewetting-induced manufacturing demonstrated here for MoS$_2$ foam is compatible with a wide variety of "2D inks". The approach thus enables reduced graphene oxide (rGO)[42] and titanium carbide (Ti$_3$C$_2$T$_x$, metallically conductive MXene)[44], to be easily printed into 3D architected structures with precise control over structural hierarchy, multiscale porosity, conductive pathways, and spatial connectivity as shown in Supplementary Fig. 18. Additionally, our methodology represents an apparent nexus to merge emerging concepts in 2D layered materials, such as contact resistance[45], Janus and van der Waals (vdW) heterostructure[46], phases engineering[47] and defect tailoring[45,48,49]. at successive length scales, producing new interfaces and properties not seen in either bulk materials or atomically thin nanosheets. Our work thus shall have profound implications for potentially enabling applications beyond Li-ion storage and integrating materials beyond MoS$_2$.

## Methods

### Preparation of chemically exfoliated MoS$_2$
For chemically exfoliated MoS$_2$ (ce-MoS$_2$), Li-intercalation was accomplished by immersing 2 g of MoS$_2$ powder in 15 ml of 0.8 M n-butyl Li in hexane. The mixture was stirred vigorously in an Ar-filled glovebox for 96 h. The slurry was filtered over Whatman filter paper (#41, ashless) and rinsed with 300 mL hexane. Next, 150 mL of deionized water (DI-H$_2$O) was added to the intercalated compound. And the mixture was sonicated to yield exfoliated monolayers. After ultrasonication, the exfoliated sheets were repeatedly washed over Millipore (pore size 200 nm) filter paper. The resulting monolayer 2D ce-MoS$_2$ sheets were resuspended to 250 μg ml$^{-1}$ in a mixture of DI-H$_2$O and IPA (7:3, v/v) for ink for the following electrohydrodynamic (EHD) printing.

### Dewetting-induced manufacturing
Experiments were performed using a customized EHD printing setup. Copper (Cu) foils were cleaned by sonication in ethanol for 20 min. The solution of ce-MoS$_2$ (250 μg mL$^{-1}$ in a mixture of DI-H$_2$O and IPA (7:3, v/v)) was fed to the spinneret (gauge 23 TW needle) by a programmable syringe pump. An external electric field of 0.75 kV cm$^{-1}$ was generated with a high-power supply (ES 40P-20 W/DAM, Gamma high voltage research). The flow rate was carefully maintained at 7 μL min$^{-1}$. The aerial mass loading of deposited MoS$_2$ foam hinges on the deposition time. A high-speed camera was implemented to observe and adjust the flow rate quickly. Note that no binders, additives, or conductive paste were used.

### Characterizations
A ZEISS ULTRA-55 scanning electron microscopy (SEM) equipped with a Quantax EDX (Xflash® 6|100) was utilized to provide morphological views operating at 5 kV. Raman spectra were collected using a Witec alpha 300 confocal Raman microscope equipped with a RayShield coupler. A 473-nm solid-state laser as the excitation source. The excitation light with a power of 2.5 mW was focused onto the sample by a 100X objective lens (N.A. = 0.9). The signal was collected by the same objective lens, analyzed by a 0.75-m monochromator, and detected by a liquid-nitrogen-cooled CCD camera. HR-TEM imaging was conducted using a Thermofisher USA (former FEI) Titan Themis Z transmission electron microscope (TEM) equipped with a double Cs (spherical aberration) corrector operating at 300 kV.

### Mechanical measurement
Nanoindentation experiments were conducted on a Bruker Hysitron TI 950 Premier nanomechanical test platform. A Berkovich diamond nanoindenter Xprobe 2D with an included angle of 142.35° and a radius of 150 nm was used to locate and image the 3D architected MoS$_2$ and

perform the indentation test. The program had three steps, (1) pressing samples for 10 s, (b) holding for 10 s, and (3) releasing the load for 10 s.

### Numerical modeling
The diffusion of Li ions was taken as a time-dependent process and was governed by Fick's second law. In COMSOL™, the module of transport of diluted species was utilized to model the diffusion process. The flux of Li ions was controlled by specifying the constant current density. We defined the full lithiation, i.e., SOC = 100%, when the structure attains the maximum theoretical capacity ($C_{max}$) of MoS$_2$. The structural Mechanics module of COMSOL™ was utilized to calculate the corresponding stress. For our case, when Li progresses, the host material experiences the elastic strain (and lithiated-induced strain ($d\varepsilon_{ij}^l$) i.e., thus the total strain becomes $d\varepsilon_{ij}^t = d\varepsilon_{ij}^e + d\varepsilon_{ij}^l$ that linearly changes with Li concentration into MoS$_2$ structure. The relation between stress and strain can be established using Hook's law as followings:

$$d\varepsilon_{ij}^e = d\left\{\frac{1}{E}\left[(1+\vartheta)\sigma_{ij} - \vartheta\sigma_{kk}\delta_{ij}\right]\right\} \tag{1}$$

where $\vartheta$ is Poisson's ratio, and $E$ is the elastic modulus of MoS$_2$, and $\sigma_{ij}$ is the evolved stress. For $i = j$, $\delta_{ij} = 1$; otherwise, $\delta_{ij} = 0$.

The bottom end of the structure was taken as fixed, while other areas were free to expand, replicating the experimental conditions of lithiation. Thermal-strain approach was utilized to calculate the evolved stress during Li progression, i.e., an arbitrary thermal expansion coefficient equivalent to the partial molar volume of MoS$_2$ was incorporated. For instance, $d\varepsilon_{ij}^l = \alpha\triangle T\delta_{ij}$, where $\alpha$ is equivalent to partial molar volume and $\triangle T$ represents the $\triangle c$. The diffusion coefficient, partial molar volume (molar mass/density), $C_{max}$, $E$ (elastic modulus) and $\vartheta$ (Poisson's ratio) were taken to be as $9 \times 10^{-16}$ m$^2$ s$^{-1}$, $3.16 \times 10^{-5}$ mol m$^{-3}$, 31610 mol m$^{-3}$, 2 GPa and 0.125, respectively. The structure parameters were comparable to the sizes taken by SEM images of the fabricated structure. As the structure experiences the huge volume expansion, the non-linear or large deformation was employed in COMSOL™. The element size of the mesh was taken small enough to ensure the solution convergence.

### Electrochemical characterization
Anodes were directly printed on a Cu current collector without additional processing steps. The loading level of spatially homogeneously MoS$_2$ foam was ~1, 2.2, and 3.5 mg cm$^{-2}$, respectively, and in the cases of reference MoS$_2$ crumples, wrinkled films and were ~1 mg cm$^{-2}$, respectively. CR2032 coin-type cells were assembled in an argon-filled glove box containing pure Li metal foil as the counter electrode. The electrolyte was 1.0 M Lithium hexafluorophosphate (LiPF$_6$) in a mixture of ethylene carbonate (EC) and dimethyl carbonate (DMC) with a volume ratio of 1:1 (Sigma-Aldrich). Advantec GC50 microporous glass fiber with a thickness of 19 μm was used as a separator. Galvanostatic cycling was conducted on a computer-controlled Neware battery test system at different current densities (0.2, 0.5, 1.0, 2.0, 5.0, and 10 A g$^{-1}$) in a potential range of 0.01–3.0 V vs. Li/Li$^+$. The cyclic voltammetry (CV) tests were carried out to examine the electrode reaction under the scan rate of 1, 2, 5, and 10 mV s$^{-1}$ with a potential range of 0.01–3.0 V vs. Li/Li$^+$. Electrochemical impedance spectra (EIS) were recorded in a frequency range from 10$^6$ to 0.01 Hz. At the same time, the disturbance amplitude was 5 mV. CV and EIS were conducted with a Biologic VMP3 electrochemical workstation. All electrochemical measurements were under constant 25 °C. As for the volumetric capacity calculation, the synthesized MoS$_2$ foam anode (thickness of 10 μm and mass loading of 1.1 mg) was deposited on the Cu substrate (diameter of 1.13 cm). The anode is binder-free and carbon-free. Active material weight percentage is 100%. The overall packing density is 1.05 ± 0.05 g cm$^{-3}$. Then

the current density and areal capacity based on the mass loading were transferred into the areal current density and volumetric capacity.

## MoS$_2$ foam // LiFePO$_4$ full cell measurement

MoS$_2$ foam works as the anode in the full cell LIB configuration, while the commercial LFP (lithium iron phosphate, LiFePO$_4$) worked as the cathode in a 2032 coin-type cell. Electrolyte is 1 M LiPF$_6$ in EC:DMC 1:1 solution. The cathodes consisted of 90 wt% LiFePO$_4$, 5 wt% polyvinylidene fluoride (PVDF), and 5 wt% super-P on Al current collectors. The anode and cathode active materials loading was adjusted to ensure that MoS$_2$ foam anode capacity was 10% higher than the LiFePO$_4$ cathode. The active material loading was 1 mg cm$^{-2}$ for the MoS$_2$ foam anode and 10 mg cm$^{-2}$ for the LFP cathode. Pre-activation of electrodes were conducted in half-cell. MoS$_2$ foam/Li metal half-cell was charged/discharged (0.01−3 V vs. Li/Li$^+$) at 0.5 A g$^{-1}$ for 2 cycles and at 0.2 A g$^{-1}$ for 2 cycles. Then the half-cell was discharged to 0.01 V at 0.1 A g$^{-1}$ and continued discharging to 0.01 V at 0.05 A g$^{-1}$ and 0.02 A g$^{-1}$. The pre-activation of cathode LFP in LFP/Li metal half-cell followed the similar process within the potential window of 3.2−3.6 V vs. Li/Li$^+$. It charged/discharged (3.2−3.6 V vs. Li/Li$^+$) at 0.5 A g$^{-1}$ for 2 cycles and at 0.2 A g$^{-1}$ for 2 cycles. Then the half-cell charged to 3.6 V vs. Li/Li$^+$ at 0.1 A g$^{-1}$ and continued charging to 3.6 V vs. Li/Li$^+$ at 0.05 A g$^{-1}$ and 0.02 A g$^{-1}$. After pre-activation, both half cells were disassembled and the MoS$_2$ foam anode and LFP cathode were reassembled into a full cell. Each full cell was aged 24 h at room temperature before commencing the electrochemical tests. The specific capacity of the full cell is calculated based on the mass of the MoS$_2$ anode electrode. Galvanostatic cycling was conducted on a computer-controlled Neware battery test system at different current densities (0.2, 0.5, 1.0, 2.0, 5.0, and 10 A g$^{-1}$) in a voltage range of 0.6−3.6 V. A Biologic VMP3 electrochemical workstation carried out the CV tests to examine the electrode reaction under the scan rate of 0.1 mV s$^{-1}$ with a voltage range of 1−4.2 V. All electrochemical measurements were taken at 25 °C. Energy density calculation is based on the total mass of MoS$_2$ foam (1 mg) and LFP (10 mg). The energy density at 0.2 A g$^{-1}$ current density is calculated through the equation Energy density = energy/(mass of anode + mass of cathode), i.e. 0.0027 Wh/(1 + 10) mg = 245 Wh kg$^{-1}$.

## X-ray absorption near edge structure (XANES)

The *operando* Mo K-edge X-ray absorption near edge structure (XANES) was measured through a punch cell with a hole covered with Kapton using the fluorescent mode at beamline TPS44A at National Synchrotron Radiation Research Center (NSRRC) in Taiwan. The XANES spectra were calibrated and normalized by the Athena (version 0.8.056)/IFEFFIT (version 1.2.11) software.

## Data availability

The data from this study are available from the corresponding author on reasonable request.

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

## Acknowledgements

V.T. and X.W. are indebted to the support from the King Abdullah University of Science and Technology (KAUST) Office of Sponsored Research (OSR) under Award No: OSR-2018-CARF/CCF-3079. V.T. acknowledges the support from Saudi Aramco, KAUST Catalysis Center (KCC), and KAUST Solar Center (KSC). H.-Y.C. and C.-C.L. acknowledge the support from 2030 Cross-Generation Young Scholars Program of the National Science and Technology Council in Taiwan under grant no. MOST 111-2628-E-007-018. H.-Y.C. and C.-C.L. thank Dr. Jyh-Fu Lee, Dr. Chih-Wen Pao, and Dr. Jeng-Lung Chen for help with the operando synchrotron XAS at the National Synchrotron Radiation Research Center, Taiwan. V.T. and X.W. are grateful to Dr. Bin Yao, and Prof. Yat Li for their help in the capacitive mechanism study, as well as Dr. Dan Sun and Prof. Elton Cairns for their support and initial discussion of electrochemical characterization and preparation.

## Author contributions

X.W. and V.T. conceived and designed the project. X.W., C.W., Y.C., J.-H.F., Z.Y., and Y.-H.C. performed the synthesis of ce-MoS$_2$, rGO, and MXene. X.W., C.-C.L., K.Q., Y.-H.C., M.B.H., and W.Z. carried out the electrochemical characterizations. L.D., O.S.A., and W.X., reviewed electrohydrodynamic printing. A.Y.L. and J.K. measured the strain-doping profile. X.W. and C.-C.L. calculated the pseudo-capacitive process. N.Q. performed the numerical modeling. All the authors discussed and contributed to the results. J.K., H.-Y.C., and V.T. wrote the paper.

## Competing interests

The authors declare no competing interests.

## Additional information

[1]Physical Science and Engineering Division, King Abdullah University of Science and Technology, Thuwal 23955-6900, Saudi Arabia. [2]Department of Materials Science and Engineering, National Tsing Hua University, Hsinchu 300, Taiwan. [3]Molecular Foundry, Lawrence Berkeley National Lab, Berkeley, California 94720, USA. [4]Department of Electrical Engineering, Massachusetts Institute of Technology, Cambridge, Massachusetts 02139, USA. [5]Department of Chemical System Engineering, School of Engineering, The University of Tokyo, Tokyo 113-8656, Japan. [6]Saudi Aramco, Chemicals R&D Lab at KAUST, Research and Development Center, Thuwal 23955-6900, Saudi Arabia. [7]Guangdong Provincial Key Laboratory of Plant Resources Biorefinery, School of Chemical Engineering and Light Industry, Guangdong University of Technology (GDUT), 100 Waihuan Xi Road, Panyu District, Guangzhou 510006, China. ✉e-mail: hanyi.chen@mx.nthu.edu.tw; vincent@g.ecc.u-tokyo.ac.jp

