## [Peer Review File · Nature Communications]

Three-dimensional Hierarchically Porous MoS₂ Foam as High-Rate and Stable Lithium-ion Battery AnodeREVIEWER COMMENTS

Reviewer #1 (Remarks to the Author):

In this manuscript, the authors have reported the dewetting-induced assembly of 2D MoS₂ nanosheets into a 3D foam with structural hierarchy across seven orders of magnitude, and explore the mechanism of extraordinary electrochemical performance by pseudocapacitive charge storage and in-situ XANES. This work suggests an advanced way of developing optimized 3D architected materials for lithium storage. Therefore, I recommend this manuscript to be accepted after the following revisions:

1. The delithiation potential of MoS₂ foam is around 1.2 V, which is too high for an anode material and would result in lower voltage and energy density for the full-cells compared with the state-of-the-art carbonaceous anode materials. So where is the application prospect of this material?
2. I am not sure why the authors observed the two highly reversible redox reaction couples (peak 1&4, peak2&3) in MoS₂ foam. What does the Peak 5 represent?
3. In the figures of capacitive and diffusion-controlled charge storage contributions (Figure 4d and Figure S1), the authors need to show the complete data. And the Figure S10 does not appear in the manuscript.
4. In Figure 3g, how to determine the uniform and porous SEI formed on its surface? All of them seem to exhibit crystalline components of electrolyte salts that are not completely washed. The same phenomenon is also observed in Figure S8. In my opinion, the high magnification TEM or HAADF images should be added to further illustrate the distribution, structure and composition of SEI.
5. The specific fitted values of Ohmic (R_s), solid electrolyte interface (R_{SEI}), and charge transfer resistance (R_{ct}) need to be shown for a more effective comparison.
6. The descriptions in many conclusions of this work are too brief, and there is no display and description of core data. Such as Li⁺ diffusion coefficients, the peaks in the CV, EIS, and so on.
7. There is a serious problem with the format of the reference.

Reviewer #2 (Remarks to the Author):

This paper presents an interesting idea of 3D hierarchically structured MoS₂ by using electrohydrodynamic printing and its use in Li-ion battery. The fabricated MoS₂ foam is claimed to be interpenetrating network and mechanical resilient, in addition to have a high charge transport and ion diffusivity property. The Li-ion battery based on 3D porous MoS₂ anode assembled with LiFePO₄ cathode is claimed to be shown an extraordinary electrochemical performance outperforming most reported MoS₂ based LIB anodes and state-of-the art materials. However, there are several concerns described below prevent this manuscript to be accepted for publication:

1. The novelty of the work throughout the manuscript is not strong. The similar manufacturing process and setup adopted by the authors in their previous publication (10.1002/adma.201703863) for the preparation of structurally deformed MoS₂. Furthermore, the authors claim that electrochemical performance of 3D foam MoS₂ is exceeding the current state of art of other materials (line 57-58: "More importantly, their electrochemical performances remain inferior to those of BP, Si, Si-graphene, or graphene benchmarks"). Here, authors are not comparing properly the electrochemical high-rate and stable performance with current state of art of anode material (e.g. Ming Chen et al. (10.1007/s12274-020-3142-9) reported the reversible capacities of P-doped Si@C electrode.
2. The stability of the fabricated MoS₂ foam in terms of physical and chemical stability need to be analyzed systematically. For example, authors used MoS₂ foam without any conducting element (e.g. carbon), what was the electron conductivity of porous MoS₂ foam? How, the conductivity of bare MoS₂ foam able to reach the required conductivity for electron flow? Did authors study the effect of Li-intercalation and its effect on electron conductivity? Did authors study post characterization of the MoS₂ foam and verify the stability of MoS₂ maintaining the 3D porous structure? What caused cell failure of these batteries?

3. The pore distribution and pore connectivity are well presented. However, once electrolyte is depleted, how do authors propose re-distribution of electrolytes? High surface area of fabricated MoS₂ foam means much contact area with the electrolyte, leading to an increased formation of solid electrolyte interface (SEI) on the MoS₂ surface. This formation consumes Li-ions irreversibly which are assembled in the SEI layer as inorganic lithium compounds like Li oxides etc. which cannot be reused for charge/discharge of the cell. For this reason, a high surface anode material shows a capacity loss. Here authors need to investigate the composition and phase of the SEI compound form over the cycle to insure the long-term performance of the cell.

4. Authors stated that “High-resolution XPS spectra of Mo 3d prove pure 2H phase in MoS₂ foam” However, it is well known that the 2H phase in MoS₂ is semiconducting in nature and 1T phase is more metallic in nature. How the presence of 2H semiconducting phase of MoS₂ foam will justify for the electron conductivity of anode without any conducting media? As MoS₂ change the phase from a trigonal prismatic (2H-MoS₂) to an octahedral (1T-Li_xMoS₂) phase during lithiation/de-lithiation. Authors need to study systematically the lithiation and de-lithiation in MoS₂ and relevant reaction/phase change take place during the charging/discharging. After several cycles, this phase transition in MoS₂, may lead to detachment and disintegration of MoS₂ to LiS₂ and Mo nanoparticles. Therefore, authors need to test structural as well mechanical stability of MoS₂ foam over the cycles.

5. The authors achieved energy density and power density of 248 Wh/kg and 207 Wh/kg at 200 mA/g and it shows the superior compared to reference materials. It is not clearly stated that if the calculation is based on the total weight of anode or just the active material MoS₂? The detail calculation for energy density and power density needs to be provided by author to support their claim. The detail information should include all the weight and size of the component with their supplier details for the repeatability of the work.

Reviewer #3 (Remarks to the Author):

In this work, the authors skillfully prepared 3D MoS₂ foam, which can provides an interpenetrating network for efficient charge transport, rapid ion diffusion, mechanically resilient and chemically stable support for electrochemical reactions. The results are interesting, and could provide some useful information to other scientists in the related research fields. The manuscript is well organized. In overall, I would like to recommend its publication in Nature Communications after minor revisions. Here are the comments in detail:

- 1、 In Figure 1a, whether the solvent is DI-H₂O or mixture of DI-H₂O and IPA (7:3, v/v), the author should be written clearly in the experimental section.
- 2、 Has the author explored the effect of external electric field and flow rate on the material in detail?
- 3、 It is recommended to add Density Functional Theory (DFT) calculations to adequately prove the advantages of the material of MoS₂ foam.
- 4、 More close related papers should be referenced for strengthen the research background.
Example: (1) Small method, 2021, 5, 2100508. (2) Advance Science, 2022, 9, 2104504. (3) Small, 2022, 18, 2107365.
- 5、 The typos and language errors need to be checked again.

August 1, 2022

Dear Reviewers:

First, we would like to thank all the reviewers for the important comments that will help us improve the manuscript. We have addressed all the comments as explained below. All changes to the manuscript are marked in blue.

Referee #1 (Remarks to the Author):

In this manuscript, the authors have reported the dewetting-induced assembly of 2D MoS₂ nanosheets into a 3D foam with structural hierarchy across seven orders of magnitude and explore the mechanism of extraordinary electrochemical performance by pseudocapacitive charge storage and in-situ XANES. This work suggests an advanced way of developing optimized 3D architected materials for lithium storage. Therefore, I recommend this manuscript to be accepted after the following revisions:

1. The delithiation potential of MoS₂ foam is around 1.2 V, which is too high for an anode material and would result in lower voltage and energy density for the full cells compared with state-of-the-art carbonaceous anode materials. So where is the application prospect of this material?

Author reply: We agree with the reviewer that the delithiation potential of 3D architected MoS₂ foam may limit the application prospect, specifically for electric vehicles. The 3D architected MoS₂ foam holds prospects for applications that require a high-power density and long cycling life. These include wearable and implantable electric devices. In addition, with emerging high-voltage cathode materials, such as LiNi_{0.5}Mn_{1.5}O₄ (working potential is approximately 4.9 V vs. Li/Li⁺), 3D architected MoS₂ foam may find good use as the anode in a 3.6 V cell. Furthermore, the excellent rate performance of 3D architected MoS₂ foam makes it an ideal candidate for Li-ion hybrid capacitors, potentially providing higher power density than Li-ion batteries and higher energy density than supercapacitors. We have clarified these muddy points and added potential applications of 3D architected MoS₂ foam in the introduction.

2. I am not sure why the authors observed the two highly reversible redox reaction couples (peak 1&4, peak2&3) in MoS₂ foam. What does Peak 5 represent?

Author reply: We thank the reviewer for helpful comments and have included a detailed explanation to elucidate these two highly reversible reaction couples in the manuscript. Indeed, the presence of these two reduction couples points to a different reduction reaction pathway in MoS₂ foam. As indicated in **Figure S14a** and **Figure 4b**, the first cycle of CV curves emanated from the MoS₂ bulk standard demonstrates the well-studied four-electron reduction reaction ($4\text{Li}^+ + \text{MoS}_2 + 4\text{e}^- \leftrightarrow \text{Mo} + 2\text{Li}_2\text{S}$) through an insertion (0.8 V) and irreversible conversion mechanism (0.2 V). In contrast, the Li⁺ intercalation and Mo⁴⁺ reduction in 3D architected MoS₂ foam progress with at least three reductive steps. Similarly, the oxidative process in 3D architected MoS₂ foam occurs through discrete stages rather than the one-step Li₂S decomposition into S, Li⁺, and electrons ($\text{Li}_2\text{S} \leftrightarrow \text{S} + 2\text{Li}^+ + 2\text{e}^-$ at 2.5 V) in MoS₂ bulk. (*Nat. Commun.* 7, 1, 2016; *Adv. Mater.* 29, 1603020, 2017; *Adv. Mater.* 28, 9385, 2016). We further conducted Raman spectroscopy of MoS₂ foam before and after the first discharge cycle in tandem with MoS₂ foam after the first charge cycle (**Figure S14b**). We observed the formation of Li₂S after the 1st discharge cycle to 0.01V and the emergence of sulfur (S) after recharging to 3V. It is noted that the signature of Mo ions remains discernible, and the presence of LiOH is the result of the SEI formation process. In the following cycles (**Figure 4b**), two highly reversible redox reaction couples (peaks 1&4 and 2&3) emerge and contribute most of the capacity, while peak 5 (2.5V) can be ascribed to Li₂S decomposition. We observed consistent trends in charge/discharge plots (**Figure S4b**), where the conversion reaction in MoS₂ foam gives rise to Li₂S and Mo. Notably, the resultant Mo atoms are in the vicinity (possibly remain in intimate contact) of the Li₂S matrix within the hierarchical foam structure, enabling rapid charge transfer with Li⁺ ions. One Mo atom can accommodate up to six Li⁺ ions and then forms Mo/Li_x clusters. Both Mo and S, therefore, participate in the reversible redox reactions, contributing to the extremely high specific capacity of more than 1,500 mAh g⁻¹ at 1 A g⁻¹. This highlights the presence of a new charge storage kinetics in hierarchically structured MoS₂ foam that is not observed in pristine MoS₂ bulk or stacked MoS₂ nanosheets (the closest one being *Energy Storage*

Materials, 16, 37, 2019). We are currently employing theoretical calculations to substantiate our observation further and will report a comprehensive study in our following report.

Figure 4b | CV measurements feature the 2nd cycle of MoS₂ foam and MoS₂ bulk electrode under 1 mV/s in the voltage window of 0.01–3 V.

Figure S14 | Electrochemical reaction of MoS₂ foam during the 1st cycle. (a) The 1st cycle CV profiles of MoS₂ foam and MoS₂ bulk electrode measured at 1 mV/s in the voltage window of 0.01–3 V. (b) Ex-situ Raman comparison of pristine, 1st discharge, and 1st charge MoS₂ foam.

3. In the figures of capacitive and diffusion-controlled charge storage contributions (Figure 4d and Figure S1), the authors need to show the complete data. And Figure S10 does not appear in the manuscript.

Author reply: We have now included **Figures 4c-d** and **S15** in the revised manuscript to comprehensively reveal the capacitive and diffusion-controlled charge storage contributions of all samples, including reference MoS₂ bulk and MoS₂ wrinkled films, MoS₂ crumples, and MoS₂ foam, respectively. The capacitive contributions are quantified by collecting CV curves at scan rates of 1, 2, 5, and 10 mV s⁻¹ over the voltage range from 0.5 to 2.5 V, the following equation: $i = k_1 v + k_2 v^{1/2}$, where $k_1 v$ and $k_2 v^{1/2}$ correspond to capacitive and diffusion contributions to the measured current, respectively. v is the scan rate (mV s⁻¹). The pseudocapacitive

contributions for MoS₂ foam, crumples, wrinkled films, and bulk are 82.8%, 34.2%, 12.2%, and 0%, respectively, in good agreement with the calculated *b*-values. Accordingly, the capacitive behavior begins to take hold in wrinkled films because of increased surface area. The capacitive energy storage contribution becomes more pronounced in highly porous crumples because of the large surface area and highly defective basal plane. It is known that these properties are highly associated with oxidation and reduction reactions. However, the limited contacts between MoS₂ crumples constrict the electron transport and therefore give rise to inferior pseudocapacitive contribution to that of MoS₂ foam. Notably, in MoS₂ foam, the ratio of the pseudocapacitive storage to the total charge storage reaches 99% at 5 mV s⁻¹ (**Figure S16**). The predominate pseudocapacitive storage again highlights the unique combination of the easily accessible surface-redox active sites, the efficient interconnection, and the shorter charge transfer distance from the surface to deep inside the foam structure.

In addition, we have now included **Figure S10**, which features the Raman and XPS characterizations of MoS₂ foam before and after scanning for 1,000 cycles.

4. In Figure 3g, how to determine the uniform and porous SEI formed on its surface? They all seem to exhibit crystalline components of electrolyte salts that are not thoroughly washed. The same phenomenon is also observed in Figure S8. In my opinion, the high magnification TEM or HAADF images should be added to further illustrate SEI's distribution, structure and composition.

Author reply: Indeed, the HRTEM or HAADF images shall elucidate the spatial distribution, structure, and composition of SEI. To this end, we have included a series of SEM (**Figures 3f-g** and **S8**), STEM, and HRTEM (**Figure S9**) images of the SEI deposited along the framework of MoS₂ foam. From SEM, the SEI layer forms a conformal coating along the hierarchical framework of MoS₂ foam after 1,000 charge/discharge cycles. The large pore size coupled with robust yet responsive struts prevents aggregation and restacking of individual nanosheets, creating room for the SEI and thus the recoverability to remedy the volumetric expansion. Meanwhile, the hierarchically porous structure with highly interconnected micro- and nano-channels enables highly efficient ion transport throughout the foam, reaching the innermost pores, thus guaranteeing continuous Li diffusion and outstanding reversibility. In stark contrast, the SEI can be seen to rampantly deposit all over the reference samples (**Figure S8**). TEM, STEM, and corresponding EDX analysis of the SEI layer (**Figure S9**) further corroborate the observation in SEM. A thin layer of SEI (about 20 nm thickness) uniformly covers the contour of MoS₂ foam. The boundary between MoS₂ and SEI layer (represented by C in red and O in green) can be determined by the corresponding EDX mapping and compositional analysis in **Table S4**. Note that Li salts have been removed thoroughly, as evident by the vanished signal of P and F.

Figure S9 | (a) TEM, (b) STEM images, and (c-e) corresponding EDX mapping of post-cycling of MoS₂ foam deposited with SEI layer.

Solid Electrolyte Interphase (SEI)					Electrodes	
Z	Element	Family	Atomic fraction (%)	Atomic error (%)	Atomic fraction (%)	Atomic error (%)
6	C	K	75.23	20.35	30.49	5.51
8	O	K	20.37	7.11	53.4	14.28
9	F	K	1.3	1.48	1.32	0.55
15	P	K	0.36	0.82	0.72	0.31
16	S	K	1.5	2.56	7.46	2.16
42	Mo	K	1.25	2.35	6.61	1.59

Table S4 | SEI and electrode composition analyses derived from **Figure S9**.

5. The specific fitted values of Ohmic (R_s), solid electrolyte interface (R_{SEI}), and charge transfer resistance (R_{ct}) need to be shown for a more effective comparison.

Author reply: We have added more details about specific fitted values of R_s , R_{SEI} , and R_{ct} based on the reviewer's suggestion. We analyze the intrinsic conductivity and electrochemical polarization of MoS_2 in bulk, wrinkled films, crumples, and foam through electrochemical impedance spectroscopy (EIS). The Nyquist plots of the impedance spectrum consist of two semicircles in the high-frequency region and a straight line in the low-frequency region. The intercept of the semicircle with the Re (Z) axis in the high-frequency region presents the R_s of the entire cell; the diameters of the first and second semicircle are associated with the resistance of Li-ion migration through the R_{SEI} and the R_{ct} , respectively; the slope of the straight line is related to ion diffusion efficiency. (*ACS Appl. Mater. Interfaces* 8, 19456, 2016).

Figure S12 | EIS analysis of MoS_2 bulk, wrinkled films, crumples, and foam. (a) The equivalent circuit model of MoS_2 Foam electrode. (b) Nyquist plots of MoS_2 Foam electrode at fully discharged state after 10 cycles at 100 mA g^{-1} . (c) Values of R_e , R_{SEI} , C_{SEI} , R_{ct} , and C_{dl} are obtained by fitting data to (a).

As shown in **Figure S12**, the Nyquist plots demonstrate a decrease in both the series resistance and charge-values. Additionally, the straight-line slope for MoS₂ foam is steeper than the reference MoS₂ bulk at a low frequency, suggesting more surface capacitance and more efficient ion diffusion because of its shorter ion transmission paths. The result is the enhanced ions storage capacity and reaction kinetics. (*Adv. Funct. Mater.* 30, 2005663, 2020). Meanwhile, the well-organized foam structure significantly facilitates the formation of highly conformal and spatially distributed SEI layers (**Figure 3g**) that guarantee electrochemical stability during charging and discharging transfer resistance for MoS₂ foam, while MoS₂ bulk and wrinkled films illustrate an increase in these resistance

6. The descriptions in many conclusions of this work are too brief, and there is no display and description of core data. Such as Li⁺ diffusion coefficients, the peaks in the CV, EIS, and so on.

Author reply: We apologize for the omission and have included experimental details and discussions (**Figure S9-S14**) to substantiate our claims further.

7. There is a serious problem with the format of the reference.

Author reply: We have picked up the slack and reorganized the reference to comply with the Nature Communication requirements.

Referee #2 (Remarks to the Author):

This paper presents an interesting idea of 3D hierarchically structured MoS₂ by using electrohydrodynamic printing and its use in Li-ion batteries. The fabricated MoS₂ foam is claimed to be interpenetrating network and mechanical resilient, in addition to having a high charge transport and ion diffusivity property. The Li-ion battery based on a 3D porous MoS₂ anode assembled with LiFePO₄ cathode is claimed to be shown an extraordinary electrochemical performance outperforming most reported MoS₂-based LIB anodes and state-of-the-art materials. However, there are several concerns described below that prevent this manuscript to be accepted for publication:

1. The novelty of the work throughout the manuscript is not strong. The similar manufacturing process and setup adopted by the authors in their previous publication (10.1002/adma.201703863) for the preparation of structurally deformed MoS₂. Furthermore, the authors claim that the electrochemical performance of 3D foam MoS₂ is exceeding the current state-of-the-art of other materials (lines 57-58: “More importantly, their electrochemical performances remain inferior to those of BP, Si, Si-graphene, or graphene benchmarks”). Here, the authors are not comparing properly the electrochemical high-rate and stable performance with the current state of the art of anode material (e.g., Ming Chen et al. (10.1007/s12274-020-3142-9) reported the reversible capacities of P-doped Si@C electrode.

Author reply: We regret that our initial submission did not convincingly capture the core novelty of 3D MoS₂ foam driven by the EHD printing. Our previous report entitled “Structurally Deformed MoS₂ for Electrochemically Stable, Thermally Resistant, and Highly Efficient Hydrogen Evolution Reaction (HER)” indeed lays a solid foundation for the creation of 3D MoS₂ foam but is quite independent due to the difference in morphological evolution— **randomly distributed and discrete crumples with hierarchically strained conformational elements**, including facets, folds, ridges, vertices, and wrinkles **vs. spatially ordered and coherent foam with hierarchically porous structural features**, such as vortical truss unit cell, nanopores, and struts, intertwined MoS₂ sheets, tears and holes on the basal plane, and S vacancies. Furthermore, because the MoS₂ crumples are inherently porous and discontinuous, the limited contacts between neighboring crumples serve as constriction points, thus diminishing electron accessibility and adversely limiting the prospects for integration in many applications with constrained device architectures. We have since made efforts to better showcase the importance and relevance to the next-generation manufacturing process of 3D architected foam and beyond.

Meanwhile, from a structural point of view, transition metal dichalcogenides (TMDs), which represent three-atom-thick 2D building blocks, have an intrinsic hierarchy of structure features, such as phase heterojunctions, grain boundaries between crystalline domains of sizes ranging from millimeters down to micrometers, dislocations at the nanoscale, and point defects such as S vacancies on the atomic scale. Despite the presence of innate structural hierarchies in tandem with the earth abundance and ease of solution processability, the development of 3D architected TMDs remains at an early stage compared to their metallic, inorganic, or even organic counterparts. Current manufacturing routes usually give rise to simple geometries of mesoporous and fractal-like features that recur only within two orders of magnitude, thus preventing researchers from combining TMDs' intrinsically attractive features with desirable material properties that are extrinsic to them. Indeed, the field of architected materials has been almost exclusively focused on metallic and inorganic materials, and the detailed mechanistic insights have shed light on many guidelines for inducing the "stronger-yet-ductile", "lightweight-and-flaw-tolerable", "electrochemically reconfigurable", and "brittle-to-ductile" transitions. Developing such understanding for 3D architected TMDs shall open new inroads for various applications where new properties and functionalities arise from the deliberate, multi-scale architecting of 2D atomic crystals.

From an assembly perspective, the design of 3D architected foam consists of staggering 2D MoS₂ nanosheets with bond-free van der Waals (vdW) interfaces. These interfaces feature sliding and rotation degrees of freedom among the staggered nanosheets, endowing mechanical recovery and adaptability while retaining charge storage capability. Specifically, without the constraint of chemical bonding, such a 3D vdW architected foam offers a unique combination of mechanical resilience and response against electrochemical-mechanical fatigue. When deformed, the bond-free vdW interfaces enable 2D MoS₂ nanosheets to slide or rotate against each other, providing additional pathways to accommodate the continuously dynamic cycles of tension and compression. It is noted that while such vdW interfaces have recently been demonstrated in thin film formats to endow exceptional malleability and adaptability to irregular surface topographies, this is the first account of the 3D freestanding, additive-free architecture with vdW interfaces. A percolating network of nanochannels simultaneously ensures efficient electron and ion transport and helps withstand the mechanical stress while facilitating the uniform SEI formation during the repeated electrochemical cycles. The result is the active anode that combines high specific capacities typical of batteries and the cycling stability of capacitors. We have now included a revised **Figure 3** to mechanically verify (**Figures 3a-c**), theoretically model (**Figure 3e**), and electrochemically demonstrate (**Figure 3f-g**) the abovementioned figure merits of 3D architected with vdW interfaces. Finally, we have included the P-doped Si@C electrode in **Table 3**. We agree with the reviewer that such a claim may likely raise concerns and cause confusion. Thus, we have revised the table and toned down the text to make the comparison fair and clear.

We hope that the inclusion of these new findings allows us to better capture the full values of the EHD-printed 3D architected foam for battery applications and beyond.

Figure 3 | Structural stability and outstanding capacity retention of MoS₂ foam. (a) SEM images of pre- (left panel) and post-compression (right panel) of MoS₂ foam to 50% of displacement demonstrate an excellent recovery behavior. (b) The load and displacement curve (displacement to 50%) displays a ductile-like feature with the continuous serrated flow (gray arrows), attesting to the multistep deformation of the hierarchical structure. (c) The load and displacement curve (displacement to 10%) exhibits a resilient feature with outstanding recoverability. (d) Li-ion diffusion and the associated concentration distribution within the architected MoS₂ at different state-of-charge (SOC). The highest concentration of scale bar at SOC=100% is calculated from the theoretical capacity of pristine MoS₂. (e) Volume expansion at various SOC. About 70% at SOC=100%. SEM images of (f) pristine MoS₂ foam and (g) MoS₂ foam after 1000 charge/discharge cycles prove the uniform formation of the SEI layer while the hierarchical structure remains intact.

2. The stability of the fabricated MoS₂ foam in terms of physical and chemical stability needs to be analyzed systematically. For example, the authors used MoS₂ foam without any conducting element (e.g., carbon), what was the electron conductivity of porous MoS₂ foam? How is the conductivity of bare MoS₂ foam able to reach the required conductivity for electron flow? Did the authors study the effect of Li-intercalation and its effect on electron conductivity? Did the authors study the post characterization of the MoS₂ foam and verify the stability of MoS₂ maintaining the 3D porous structure? What caused the cell failure of these batteries?

Author reply: Following the suggestions of the reviewer, we have now added a systematic study of the electron conductivity of porous MoS₂ foam. EIS data shown in **Figure 4a** indicate significantly reduced R_{ct} and R_{SEI} of

MoS₂ foam than those of MoS₂ bulk. This is quite surprising given that both MoS₂ foam and bulk are characterized by a 2H phase crystal structure where its metal atoms are coordinated in a trigonal prismatic manner. The result is the semi-insulator with a sizeable bandgap of ~ 1.9 eV, an inferior electrical conductivity of ~ 0.03 S/m, a relatively high diffusion barrier (0.42 eV) compared to BP (0.08 eV), silicon (Si, 0.37-0.54 eV) and graphite (0.37 eV), and therefore not immediately attractive as an electrode material for any metal-ion battery. Yet, recent reports indicate that local strain in mono- or few-layer MoS₂ nanosheets can reduce its bandgap and thus enhance its electron conductivity and electrochemical activity (*Nano Lett.* 13, 5361, 2013 and *Sci. Rep.* 8, 2079, 2018). To this end, we prepared a 2-inch Cu substrate printed with continuous MoS₂ films with dissimilar morphologies, including crumples (top), architected MoS₂ (middle), and wrinkles (bottom right), made possible by the programable printing. The strain-charge doping (ϵ - n) map derived from the linear relationship between biaxial strain/charge doping and Raman shifts provides an index for quantifying the strain load (characterized to shift by ~ 1.7 cm⁻¹ per % strain) and surface electron densities. Note that $\Delta\epsilon > 0$ is indicative of tensile strain and Δn can be used to compare the relative electron densities (*Appl. Phys. Lett.* 111, 143106, 2017).

[Unpublished results] | (Left) Optical image of a 2-inch Cu substrate printed with conformationally dissimilar MoS₂ made possible by the programable, dewetting-driven destabilization assisted printing. (Right) Raman-derived strain-charge doping (ϵ - n) map.

3D architected MoS₂ foam is substantially strained ($\sim 1.75 \pm 0.15\%$ vs. $3.2 \pm 0.37\%$ of tensile strain in crumples, based on the redshift magnitudes of the Raman E_{2g} and A_{1g} peaks in **Figure s2f**) and displays a relatively higher electron density than that of the wrinkled counterpart. These results agree well with the previous reports and have profound implications on activating the 3D architected MoS₂ with the significantly decreased ion diffusion barrier with ~ 0.2 eV for Li and greatly improved conductivity of 4.66 S/m compared to that of pristine 2H-MoS₂ bulk (0.42 eV for Li-ion diffusion barrier and conductivity of 0.0576 S/m). In addition, strain-induced upshift of Mo *d* states towards the Fermi level gives rise to a more robust interaction with metal ions, indicating that the storage capacity could be directly tailored at the atomic level. Consequently, the inherently strained structure of 3D architected MoS₂ foam opens new inroads to manipulate the intrinsic activities of 2D MoS₂ building blocks, such as diffusion barrier, adsorption, and conductivity. Meanwhile, electrochemically driven dimensional changes in the anodes made of randomly restacked MoS₂ sheets lead to mechanical stress buildup at a charge-discharge current density of >5 A/g. The result is fatigue and capacity fading after only a few tenths of cycles. It is therefore widely deemed impractical, and only a handful of studies have described the Li-ion storage properties of electrodes consisting of single- and multi-layered MoS₂ nanosheets. In stark contrast, the combination of manufacturing scalability, 3D hierarchically porous and spatially interconnected networks, multiscale architectural features, and strain-engineered ion diffusion barriers and conductivity suggests that 3D architected MoS₂ may be an ideal anode alternative with high-rate, high-capacity, high-mass-loading storage, and long-term cyclability. We have schematically correlated these appealing features with the formation of hierarchical structures within 3D architected MoS₂.

Scheme S1 | A schematic illustration that summarizes the figures of merit of EHD-printed 3D architected MoS₂ foam.

Post characterization: We agree with the reviewer that more characterization and mechanical tests of the post-cycling MoS₂ foam with SEI are essential to understand the underlying mechanism further. A new set of post-characterization results are now included in **Figures S10-S11**. Anodes comprised of MoS₂ foam were pre-cycled (charge to 3 V) and then characterized by the Raman and XPS spectra. In Raman, we observed the broadening and redshift of both E¹_{2g} and A¹_g peaks, indicating pronounced strain and structural deformation. In parallel, XPS results substantiate our claim that the chemical composition and phase (trigonal prismatic, e.g., 2H) of MoS₂ foam remain unchanged after cycling. Also, we extended the post-characterization to assess the structural integrity and recoverability after cyclic deformation. As suggested in **Figures 3** and **S11**, the 3D architected structure remains largely intact by virtue of the great recoverability. Taken together, these post-cycling characterizations confirm the uniform SEI layer formation over the entire 3D architected MoS₂ foam and the outstanding chemical and mechanical stabilities, ultimately resulting in much-improved capacity retention. We hope that the inclusion of these new findings allows us to better capture the full values of the EHD-printed 3D architected MoS₂ foam for energy storage applications and beyond.

Figure S10 | Post-cycling characterization of MoS₂ foam. (a) Raman and (b) X-ray photoelectron spectroscopy spectra of pristine MoS₂ foam (color in gray) and MoS₂ foam after 1000 cycles (color in orange).

Figure S11 | Mechanical test of (a) pristine MoS₂ foam and (b) post-cycling MoS₂ foam.

3. The pore distribution and pore connectivity are well presented. However, once electrolyte is depleted, how do authors propose the re-distribution of electrolytes? High surface area of fabricated MoS₂ foam means much contact area with the electrolyte, leading to an increased formation of solid electrolyte interface (SEI) on the MoS₂ surface. This formation consumes Li-ions irreversibly which are assembled in the SEI layer as inorganic lithium compounds like Li oxides etc. which cannot be reused for charge/discharge of the cell. For this reason, a high surface anode material shows a capacity loss. Here authors need to investigate the composition and phase of the SEI compound form over the cycle to insure the long-term performance of the cell.

Author reply: We appreciate the helpful suggestions from the reviewer and have included the SEM, TEM, and STEM images to verify the composition and phase SEI formed over the cycle. **Figures 3f-g** feature the spatially distributed SEI layer on top of the well-maintained, porous foam even after 1,000 charge/discharge cycles. Notably, the stable formation of the SEI layer is a testament to the unique combination of 3D architected structure, porosity, and mechanical recoverability. In parallel, the ion channels and electron pathways along the 3D porous framework ensure uninterrupted Li diffusion and stable reversibility. Meanwhile, rampant deposition of SEI can be seen in controlled anodes made of (i) randomly stacked nanosheets, (ii) wrinkled films, and (iii) crumpled particles, respectively. To determine the thickness and compositional distribution of the SEI layer on 3D architected MoS₂ foam, we conducted a systematic study using TEM, STEM, and EDX analysis, as shown in **Figure S9**. A thin layer of SEI (thickness of ~20 nm) can be found to cover the MoS₂. Specifically, the well-defined boundary between MoS₂ and SEI layer can be clearly defined through the EDX mapping and the corresponding compositional analysis (**Table S4**). We confirm Li salt's removal because of negligible P and F signals.

We followed the standard protocol to prepare the TEM samples (*Nature Energy*, 6, 487, 2021 and *Nano Lett.* 19, 5140–5148, 2019). The coin cell in the delithiated state was disassembled inside an Ar-filled glovebox and comprehensively rinsed with DMC three times to ensure the removal of residual Li salts. Then rinse the MoS₂ anode with ethanol three times. The rinsed MoS₂ foam anode was then transferred to a Teflon-sealed scintillation vial and lightly sonicated again in ethanol to remove particles from the electrode. The solution is then drop-cast onto a lacey carbon TEM grid. The sample is then transferred into liquid nitrogen outside the glovebox using a sealed container to avoid exposure to air.

Figure S9 | Formation of secondary electrolyte interphase (SEI) layer after post-cycling of MoS₂ foam. (a) TEM, (b) STEM images, and (c-e) EDX mappings, where S is in yellow, Mo is in cyan, C is in red, and O is in green, reveal the spatial and uniform deposition of SEI on MoS₂ foam. The boundary between MoS₂ and SEI layer (represented by C in red and O in green) can be determined by the corresponding EDX mapping and compositional analysis in **Table S4**.

Solid Electrolyte Interphase (SEI)					Electrodes	
Z	Element	Family	Atomic fraction (%)	Atomic error (%)	Atomic fraction (%)	Atomic error (%)
6	C	K	75.23	20.35	30.49	5.51
8	O	K	20.37	7.11	53.4	14.28
9	F	K	1.3	1.48	1.32	0.55
15	P	K	0.36	0.82	0.72	0.31
16	S	K	1.5	2.56	7.46	2.16
42	Mo	K	1.25	2.35	6.61	1.59

Table S4 | SEI and electrode composition analyses derived from **Figure S9**.

4. Authors stated that “High-resolution XPS spectra of Mo 3d prove pure 2H phase in MoS₂ foam” However, it is well known that the 2H phase in MoS₂ is semiconducting in nature and 1T phase is more metallic in nature. How the presence of the 2H semiconducting phase of MoS₂ foam will justify the electron conductivity of the anode without any conducting media? As MoS₂ changes the phase from a trigonal prismatic (2H-MoS₂) to an octahedral (1T-LixMoS₂) phase during lithiation/de-lithiation. Authors need to study systematically the lithiation and de-lithiation in MoS₂ and the relevant reaction/phase change that takes place during the charging/discharging. After several cycles, this phase transition in MoS₂ may lead to detachment and disintegration of MoS₂ to Li₂S and Mo nanoparticles. Therefore, the authors need to test the structural as well as mechanical stability of MoS₂ foam over the cycles.

Author reply: We appreciate the helpful suggestions from the reviewer and have addressed the concerns in Q2.

5. The authors achieved energy density and power density of 248 Wh/kg and 207 Wh/kg at 200 mA/g and it shows superior compared to reference materials. It is not clearly stated if the calculation is based on the total weight of the anode or just the active material MoS₂? A detailed calculation for energy density and power density

needs to be provided by the author to support their claim. The detailed information should include all the weight and size of the component with their supplier details for the repeatability of the work.

Author reply: We agree with the reviewer that detailed information should be included to avoid confusion. The anode is made of direct printing of pristine MoS₂ nanosheets into 3D hierarchical architecture on the targeted Cu substrates. Since no binders or additives are used in the ink preparation, energy density is determined based on the total mass of printed MoS₂ foam (1 mg) and cathode LFP (10 mg). The energy density at 0.1 A/g current density is 0.0027 Wh / (1+10) mg= 248 Wh/kg. Detailed information can now be found in the Supporting information. Also, we have clarified the calculation in the revised manuscript.

Referee #3 (Remarks to the Author):

In this work, the authors skillfully prepared 3D MoS₂ foam, which can provide an interpenetrating network for efficient charge transport, rapid ion diffusion, and mechanically resilient and chemically stable support for electrochemical reactions. The results are interesting and could provide some useful information to other scientists in the related research fields. The manuscript is well organized. Overall, I would like to recommend its publication in Nature Communications after minor revisions. Here are the comments in detail

1. In Figure 1a, whether the solvent is DI-H₂O or a mixture of DI-H₂O and IPA (7:3, v/v), the author should be written clearly in the experimental section.

Author reply: We have now clarified solvent use in the Supporting Information section. The monolayer ce-MoS₂ is dispersed in DI-H₂O after chemical exfoliation. Next, the aqueous dispersion of ce-MoS₂ was resuspended to 250 µg/ml in a mixture of DI-H₂O and IPA with a final volume ratio of 7:3, v/v. This is the ratio for preparing the 2D ce-MoS₂ ink for the EHD printing process.

2. Has the author explored the effect of external electric field and flow rate on the material in detail?

Author reply: We have done a comprehensive study to explore the effect of external electric field and flow rate. Under different combinations of flow rate, charge (Q), and electric field strength (E), droplets are jetted in different modes. When E and Q are low, the large droplets are jetted out from the nozzle mainly due to gravity in the “dripping mode.” Increasing the flow rate renders fluid to form a jet or stream, namely “jet mode.” In both dripping and jet modes, the large droplets or streams coalesce into thick films on the targeted substrate. On the other hand, the “cone-jet mode” emerges when E exceeds the threshold. The cone-jet mode is characterized by the formation of the Taylor-cone-shaped extrusion near the tip of the injection needle. Usually, a higher flow rate requires a higher voltage to reach a cone-jet mode. However, tilted jets or multiple jets may occur when E is too high.

The result is the over-dispersed tiny droplets, but the spray is not stable and less controllable. Particularly, the “cone-jet mode” enables the controllability and stability of extruding the ce-MoS₂ suspension into self-dispersing and electrostatically charged droplets with a very narrow distribution in diameter (diameter is about 150 nm) blended with continuous jet streams. The stable generation of cone-jet mode is critical for forming ordered 2D patterns and thus the final 3D hierarchical architecture. The optimized conditions for MoS₂ foam formation are of 0.75 kV/cm (E), 7 µL/min (flow rate) and 200°C (substrate temperature). **Figure S1** schematically illustrated the operational conditions and the formation of different structures.

Figure S1 | Schematic illustrations of different EHD modes and the resultant morphologies. (a) Commercially available MoS₂ bulk powder and ce-MoS₂ aqueous dispersion as well as their corresponding SEM images. (b) Arrays of SEM images were taken under a different combination of annealing temperature and electric field. MoS₂ wrinkled films formed under the jet stream mode while MoS₂ foam under the cone-jet (jet stream and droplets) mode. MoS₂ crumples under high temperature and electric field. Scale bars from left to right, 200 nm, 200 nm, 1 μ m, and 2 μ m, respectively.

3. It is recommended to add Density Functional Theory (DFT) calculations to adequately prove the advantages of the material of MoS₂ foam.

Author reply: Indeed, we agree with the reviewer that to achieve the maximized performance of 3D architected TMD foam, there need to be comprehensive insights into properties, from angstrom to the micrometer. DFT calculation can certainly help in this endeavor. In this light, we aim to leverage the power of DFT that can be used to predict the fundamental properties of 3D architected materials from their smallest constituents, e.g., atoms, chemical bonds between atoms, interactions between van der Waals interfaces, and unit cells. This study is still under active development and refinement. We will report the DFT calculations in our next report.

4. More close related papers should be referenced for strengthening the research background. Example: (1) *Small method*, 2021, 5, 2100508. (2) *Advanced Science*, 2022, 9, 2104504. (3) *Small*, 2022, 18, 2107365.

Author reply: We have included all these relevant references in our reference list.

5. The typos and language errors need to be checked again.

Author reply: We profusely apology the typos and grammatic errors in the previously submitted manuscript and have addressed these issues in our revised version.

Additional corrections

We have added Kai Qi and Dr. Mohamed Ben Hassine to the author list to acknowledge their immense support during TEM and STEM characterizations. Meanwhile, Jui-Han Fu and Vincent Tung have added a new affiliation.

In summary, we are grateful to receive constructive reviews from all three referees. With all their comments addressed, we hope that the manuscript is now suitable for publication in *Nature Communications*. Thank you very much for your consideration.

Sincerely yours,

Vincent Tung

Professor,
Chemical System Engineering
The University of Tokyo

Han-Yi Chen

Associate Professor
Materials Science & Engineering
National Tsing-Hua University

REVIEWER COMMENTS

Reviewer #1 (Remarks to the Author):

The authors have well revised the manuscript. It can be published now